# Understanding and Enhancing the Planning Capability of Language Models via Multi-Token Prediction

## Abstract

Large Language Models (LLMs) have achieved impressive performance across diverse tasks but continue to struggle with learning transitive relations, a cornerstone for complex planning (Balesni et al., 2024; Wang et al., 2024b). To address this issue, we investigate the Multi-Token Prediction (MTP) paradigm and its impact to transitive relation learning. We theoretically analyze the MTP paradigm using a Transformer architecture composed of a shared output head and a transfer layer. Our analysis reveals that the transfer layer gradually learns the multi-step adjacency information, which in turn enables the backbone model to capture unobserved transitive reachability relations beyond those directly present in the training data, albeit with some inevitable noise in adjacency estimation. Building on this foundation, we propose two strategies to enhance the transfer layer and overall learning quality: *Next-Token Injection (NTI)* and a *Transformer-based Transfer Layer*. Our experiments on both synthetic graphs and the Blocksworld planning benchmark validate our theoretical findings and demonstrate that the improvements significantly enhance the model's path-planning capability. These findings deepen our understanding of how Transformers with MTP learn in complex planning tasks, and provide practical strategies to overcome the transitivity bottleneck, paving the way toward structurally aware and general-purpose planning models.

## 1 Introduction

Transformer models have achieved remarkable success across natural language processing (Vaswani et al., 2017; Devlin et al., 2019; Brown et al., 2020), computer vision (Dosovitskiy et al., 2021; Carion et al., 2020; Liu et al., 2021), reinforcement learning (Parisotto et al., 2020; Chen et al., 2021a; Janner et al., 2021), program synthesis (Chen et al., 2021b; Nijkamp et al., 2023), and complex planning (Chen et al., 2021a; Lehnert et al., 2024). However, a fundamental question remains: do these models truly possess planning capabilities, or do they merely rely on reconstructing patterns from training data? This question is particularly critical in complex planning tasks, which often require compositional planning to generate coherent sequences of actions toward a goal. In such tasks, it is natural and effective to abstract the problem as path finding on a graph, where nodes represent states and edges represent executable actions. path finding not only lies at the core of many classical planning problems but also closely relates to sequential decision-making in real-world complex tasks, such as robotic motion planning, automated scheduling, and step-wise reasoning in mathematical proofs. Under this abstraction, standard autoregressive Transformers typically perform reliable planning on paths observed during training (Wang et al., 2024b).

However, the performance of these models degrades substantially when the task requires transitive planning, which demands combining information from multiple path segments to infer new reachability relations: as demonstrated by Wang et al. (2024b), standard autoregressive Transformers would fail to infer that node $A$ could reach node $C$ when the training data contain both paths from $A$ to $B$ and from $B$ to $C$ but no paths from $A$ to $C$. This limitation not only prevents the model from generalizing to unseen paths in complex planning tasks but also highlights a fundamental bottleneck of current Transformers in structured planning (Zhang et al., 2024). Therefore, understanding and improving the model's compositional learning ability is crucial for enhancing Transformers in structured planning and sequential prediction tasks.

To address this issue, we explore the Multi-Token Prediction (MTP) paradigm, in which the model predicts multiple future nodes in a single training step, providing richer supervision signals and showing potential for modeling long-range dependencies and structural relationships. MTP has been adopted in a number of leading AI companies and their models such as Meta and DeepSeek (Gloeckle et al., 2024; Liu et al., 2024), but their underlying mechanism, especially for planning, remain largely unexplored. In this work, we build on the analytical framework of ALPINE (Wang et al., 2024b) to systematically investigate the effect of MTP on path planning, and propose architectural enhancements to strengthen the ability of Transformers to learn transitive reachability relations. Our study provides both theoretical insights and practical guidance for developing future Transformers with stronger reasoning and planning capabilities.

To summarize, our contributions include: First, through a theoretical analysis on a simplified Transformer, we show how the multi-token loss simultaneously shapes the transfer matrices and backbone network weights, revealing the coupled learning dynamics among the transfer layer and the adjacency and reachability within the backbone model (Section 3). Second, based on these insights, we propose two enhancements to the architecture: (1) *Next-Token Injection (NTI)*, to explicitly inject intermediate nodes as multi-hop supervision; and (2) a *Transformer-based Transfer Layer* to maintain structural consistency across prediction steps (Section 4). Third, we conduct extensive experiments on synthetic graphs as well as the Blocksworld planning benchmark and show that these methods significantly improve prediction accuracy, and the learned transfer matrices progressively approximate the ground-truth adjacency matrices (Section 5). These findings indicate that MTP together with our enhancements provides better support for transitive relations, advancing models toward stronger structural planning capabilities.

**Additional Related Work**   Our work relates to recent studies on the structured planning ability of large language models (LLMs). Prior work has examined LLMs in task planning with external modules or used future-information augmentation methods like Multi-Token Prediction (MTP) to boost inference efficiency and empirical performance. Unlike these, we adopt a theoretical view on gradient-based learning dynamics, showing how MTP fundamentally capture long-range dependencies by shaping representations during training. See Appendix A for more details.

## 2 SETTING AND PRELIMINARIES

We use $\boldsymbol{a}$ to denote a column vector, $\boldsymbol{A}$ denotes a matrix; the $i^{th}$ component of $\boldsymbol{a}$ is written as $\boldsymbol{a}_{(i)}$; the $(i, j)$ entry of $\boldsymbol{A}$ is written $\boldsymbol{A}_{(i,j)}$; the $i$-th row (column) of $\boldsymbol{A}$ is denoted $\boldsymbol{A}_{(i,:)}$ ($\boldsymbol{A}_{(:,i)}$).

### 2.1 PATH PLANNING OVER SIMPLE DIRECTED GRAPHS WITH LANGUAGE MODEL

To evaluate the planning capability of an autoregressive language model, we construct path-planning tasks on directed graphs. Let $\mathcal{G} = (\mathcal{V}, \mathcal{E})$ be a directed acyclic graph with node set $\mathcal{V}$ and edge set $\mathcal{E}$. For any $u, v \in \mathcal{V}$, the presence of $(u, v) \in \mathcal{E}$ indicates a directed edge from $u$ to $v$.

During training, each reachable source–target pair $(s, t)$ (i.e., $t$ can be reached from $s$ via one or more edges) is encoded as a token sequence "s t s a b c t \n", where $s$ and $t$ denote the source and target, $a, b, c$ represent intermediate nodes, and \n marks sequence termination. The model learns in an autoregressive fashion by predicting every next token in turn. At test time, only the prefix "s t" is provided; the model must autoregressively complete a valid path from $s$ to $t$, respecting the graph's adjacency and reachability constraints. This procedure measures the model's ability to capture both one-step adjacency and long-range reachability information.

We denote the ground-truth adjacency matrix and reachability matrix of the graph by $\boldsymbol{A}^{\text{true}}$ and $\boldsymbol{R}^{\text{true}}$, respectively:

$$\boldsymbol{A}^{\text{true}}_{(i,k)} = \begin{cases} 1, & \text{if } (i,k) \in \mathcal{E}, \\ 0, & \text{otherwise.} \end{cases} \qquad \boldsymbol{R}^{\text{true}}_{(t,k)} = \begin{cases} 1, & \text{if there exists a path } k \to t \text{ in } \mathcal{G}, \\ 0, & \text{otherwise.} \end{cases}$$

### 2.2 HIERARCHICAL EVALUATION OF GENERALIZATION ABILITY

Given a training set $\mathcal{D}$ where $\boldsymbol{u} = (u_1, \ldots, u_N) \in \mathcal{D}$ is a valid path in the graph, with $N$ denoting the sequence length, we define the observed adjacency and reachability matrices in the observation

graph $\mathcal{G}_{\text{obs}}$:

$$\boldsymbol{A}^{\text{obs}}_{(i,k)} = \begin{cases} 1, & \text{if } \exists \, \boldsymbol{u} \in \mathcal{D}, \, n \in [3, N-1] \text{ s.t. } u_n = i, \, u_{n+1} = k, \\ 0, & \text{otherwise} \end{cases}$$

$$\boldsymbol{R}^{\text{obs}}_{(t,k)} = \begin{cases} 1, & \text{if } \exists \, \boldsymbol{u} \in \mathcal{D}, \, n \in [4, N] \text{ s.t. } u_2 = t, \, u_n = k, \\ 0, & \text{otherwise.} \end{cases}$$

Here, $\boldsymbol{R}^{\text{obs}}$ is a subset of $\boldsymbol{R}^{\text{true}}$, contains only the reachability relations directly observed in $\mathcal{D}$.

We then partition test pairs $(s, t)$ into four *degrees* based on their observed reachability in $\mathcal{G}_{\text{obs}}$: a) degree-0 if $\boldsymbol{R}^{\text{obs}}_{(t,s)} = 1$; b) degree-1 if $\boldsymbol{R}^{\text{obs}}_{(t,s)} = 0$ but there exists $u$ such that $\boldsymbol{A}^{\text{obs}}_{(s,u)} = 1$ and $\boldsymbol{R}^{\text{obs}}_{(t,u)} = 1$; c) degree-2 if it is neither degree-0 nor degree-1 but there exists $u$ such that $\boldsymbol{A}^{\text{obs}}_{(s,u)} = 1$ and $(u, t)$ is degree-1; d) degree-3 otherwise.

Following the standard architecture of Generative Pretrained Transformer (GPT) (Radford et al., 2018), each Transformer layer comprises multi-head attention (MHA), residual connections, layer normalization (LN), and a feed-forward network (FFN), as

$$\text{Transformer}(\boldsymbol{X}) = \text{FFN}\big(\text{LN}_2(\text{MHA}(\text{LN}_1(\boldsymbol{X})) + \boldsymbol{X})\big) + \text{MHA}(\text{LN}_1(\boldsymbol{X})) + \boldsymbol{X}. \tag{1}$$

A input sequence $\boldsymbol{u} = (u_1, \ldots, u_n)$ is first mapped to a sequence of corresponding embedding vectors $\boldsymbol{X}_{1:n} = (\boldsymbol{x}_1, \ldots, \boldsymbol{x}_n)$ using an embedding matrix $\boldsymbol{W}_t$. This sequence is then passed through the Transformer layers, which produce a sequence of contextualized hidden states

$$\boldsymbol{H}_{1:n} = (\boldsymbol{h}_1, \ldots, \boldsymbol{h}_n) = \text{Transformer}(\boldsymbol{X}_{1:n}). \tag{2}$$

For next-token prediction, the model uses the final hidden state $\boldsymbol{h}_n$, which corresponds to the last token in the input sequence. The predictive distribution is then given by:

$$\boldsymbol{p}(u_{n+1} \mid u_{1:n}) = \text{softmax}(\boldsymbol{W}_o \boldsymbol{h}_n).$$

The standard next-token training objective is the cross-entropy loss,

$$\mathcal{L} = -\sum_{n=1}^{N-1} \log \boldsymbol{p}(u_{n+1} \mid u_{1:n}). \tag{3}$$

Wang et al. (2024b) point out that GPT trained solely with next-token loss achieves over 90% accuracy on degree-0 and degree-1 tasks, but drops to about 60% on degree-2 tasks. The model can only learn the observed reachability matrix $\boldsymbol{R}^{\text{obs}}$, and fails to learn the complete true reachability matrix $\boldsymbol{R}^{\text{true}}$, highlighting its inability to generalize to transitive paths unseen during training.

## 3 MECHANISTIC UNDERSTANDING OF MULTI-TOKEN PREDICTION

In this paper, we investigate the mechanism of MTP in which multi-step tokens are used during learning to enhance the learning effectiveness, while inference is still done by next-token prediction. Some prior work also uses MTP at inference to accelerate generation, but it is not our focus (See Appendix A for more discussions). We use a shared output head architecture for MTP (Figure 1(b)). Separate learnable *transfer layers* map the backbone output to different target positions in parallel before decoding, enabling the model to share parameters across steps. This design enhances the interpretability of structural learning. The shared output head is denoted as $\boldsymbol{W}_o$, and the transfer layer as $\boldsymbol{W}^T$. In contrast, Meta's MTP architecture assigns an independent output head to each prediction step (Gloeckle et al., 2024) (Figure 1(a)). While flexible, this approach lacks parameter sharing, making it harder to model unified patterns across steps. The two architectures are mathematically equivalent when the transfer layers are linear. Therefore, we choose the first architecture for its better interpretability.

### 3.1 TRAINING DYNAMICS OF 2-TOKEN PREDICTION

To enable theoretical analysis on how 2-token supervision shapes structural learning, we simplify the Transformer model with several assumptions, similar to the ones in (Wang et al., 2024b). We

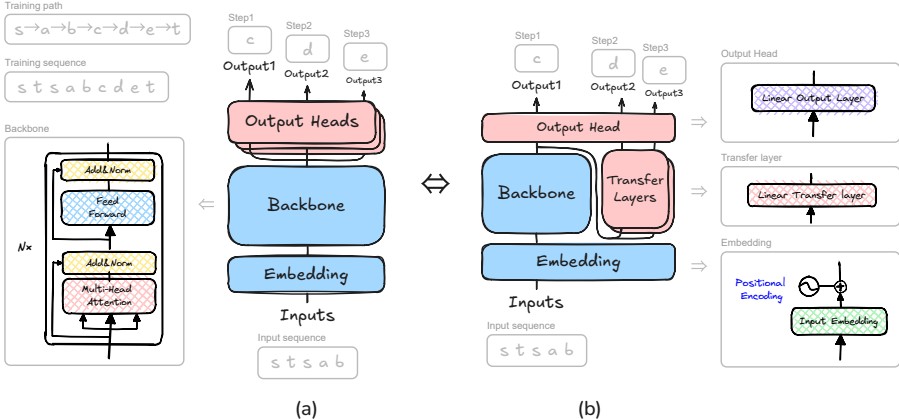

Figure 1: **Multi-Token Prediction (MTP) architectures for 3-step prediction.** (a) Meta's MTP architecture with independent output heads for each step. (b) Ours with a shared output head: i) next-step predictions are generated directly from the backbone, ii) other steps require transformation through a separate transfer layer.

consider a single-layer, single-head Transformer where positional embeddings and layer normalizations are omitted; the feed-forward network is a single linear map $\text{FFN}(\boldsymbol{X}) = \boldsymbol{X}\boldsymbol{W}^M$; and the token embedding matrix $\boldsymbol{W}_t$ and output embedding matrix $\boldsymbol{W}_o$ are identity matrices. The attention mechanism is also simplified, using a standard value projection matrix $\boldsymbol{W}^V$ but with a manually set attention matrix $\boldsymbol{\alpha}$ (replacing the standard $\text{softmax}\left(\frac{\boldsymbol{Q}\boldsymbol{K}^\top}{\sqrt{d_k}}\right)$) that restricts attention to the target token (i.e., each row is a one-hot vector with a 1 in the second column). Finally, a transfer matrix $\boldsymbol{W}^T \in \mathbb{R}^{M \times M}$ maps next-step logits to subsequent-step logits. Complete derivations under these assumptions are provided in Appendix B.

Under this setup, the hidden state at position $n$, $(\boldsymbol{H}_1)_{(n,:)}$, is the sum of feed-forward and attention outputs derived from the one-hot input matrix $\boldsymbol{U}$. Projecting this state through the output matrix $\boldsymbol{W}_o$ yields the logits:

$$(\boldsymbol{H}_1)_{(n,:)} \boldsymbol{W}_o = \left(\boldsymbol{U}\boldsymbol{W}_t\boldsymbol{W}^M + \boldsymbol{\alpha}\boldsymbol{U}\boldsymbol{W}_t\boldsymbol{W}^V\right)_{(n,:)} \boldsymbol{W}_o = \left(\boldsymbol{U}\boldsymbol{W}^M + \boldsymbol{\alpha}\boldsymbol{U}\boldsymbol{W}^V\right)_{(n,:)} = \boldsymbol{W}^M_{(u_n,:)} + \boldsymbol{W}^V_{(u_2,:)},$$

where $u_n$ is the current token and $u_2$ is the attended target token. Therefore, the logits for the next and the subsequent step are given by

$$\text{logit}_{n+1}(k) = \boldsymbol{W}^M_{(u_n,k)} + \boldsymbol{W}^V_{(u_2,k)}, \quad \text{logit}_{n+2}(k) = (\boldsymbol{W}^M\boldsymbol{W}^T)_{(u_n,k)} + (\boldsymbol{W}^V\boldsymbol{W}^T)_{(u_2,k)}.$$

For 2-Token Prediction, the objective is $\ell(\mathcal{D}) = \ell^{(1)}(\mathcal{D}) + \ell^{(2)}(\mathcal{D})$, and we focus on the second loss $\ell^{(2)}(\mathcal{D})$, which is formulated as:

$$\ell^{(2)}(\mathcal{D}) = -\sum_{u \in \mathcal{D}} \sum_{n=1}^{N-2} \sum_k \boldsymbol{U}_{(n+2,k)} \log \frac{\exp\left((\boldsymbol{W}^M\boldsymbol{W}^T)_{(u_n,k)} + (\boldsymbol{W}^V\boldsymbol{W}^T)_{(u_2,k)}\right)}{\sum_\ell \exp\left((\boldsymbol{W}^M\boldsymbol{W}^T)_{(u_n,\ell)} + (\boldsymbol{W}^V\boldsymbol{W}^T)_{(u_2,\ell)}\right)}. \quad (4)$$

Let $\hat{P}_{i,j}(k')$ be the softmax probability of predicting node $k'$ two steps ahead given current node $i$ and target $j$. Define $N_{i,j,k'}$ as the number of such occurrences in $\mathcal{D}$ and $N_{i,j} = \sum_{k'} N_{i,j,k'}$. This leads to the following theorem.

**Theorem 1.** *For any pair $(i,j)$ in the dataset $\mathcal{D}$ with $N_{i,j} > 0$, let $P^{\text{data}}_{i,j}(k') = \frac{N_{i,j,k'}}{N_{i,j}}$ denote the empirical probability of the second-next node $k'$. The contribution of this pair to the gradient $\frac{\partial \ell^{(2)}(\mathcal{D})}{\partial \boldsymbol{W}^T_{(d,k')}}$ is proportional to the prediction error $(\hat{P}_{i,j}(k') - P^{\text{data}}_{i,j}(k'))$, the sample count $N_{i,j}$, and the combined weight $(\boldsymbol{W}^M_{(i,d)} + \boldsymbol{W}^V_{(j,d)})$. When the contribution is positive, it pushes the weight $\boldsymbol{W}^T_{(d,k')}$ to **decrease**; when negative, it pushes the weight to **increase**. The total gradient is the sum of contributions over all pairs $(i,j)$ in the dataset.*

The derivations and proofs of the theorems are provided in Appendix C.1.

**Transfer Matrix Learned as an Adjacency Matrix.** Theorem 1 shows that $\boldsymbol{W}^T$ is updated by 2nd-step prediction errors. When the model underestimates the probability of reaching node $k'$ from node $i$ in two steps, the gradient increases the weight $\boldsymbol{W}^T_{(d,k')}$ from all the positive-correlated intermediate node $d$ (e.g., all the feasible $d$'s for this $i,j$ pair) to $k'$; otherwise, it decreases. Thus, if the backbone model correctly predicts the next-step node $d$, then by increasing the weight $\boldsymbol{W}^T_{(d,k')}$ to a greater extent, it will enable $\boldsymbol{W}^T$ to correctly learn the adjacency between $d$ and $k'$.

We next analyze how the 2nd-step prediction affects the backbone parameters $\boldsymbol{W}^M$ and $\boldsymbol{W}^V$ through gradients propagated from the transfer matrix.

**Theorem 2.** *For any pair $(i, j)$ in dataset $\mathcal{D}$ with $N_{i,j} > 0$, the contribution of each (current node $i$, second-step node $k'$) pair to the gradient $\frac{\partial \ell^{(2)}(\mathcal{D})}{\partial \boldsymbol{W}^V_{(j,k)}}$ is proportional to the prediction error $(\widehat{P}_{i,j}(k') - P^{\mathrm{data}}_{i,j}(k'))$, the sample count $N_{i,j}$, and the weight $\boldsymbol{W}^T_{(k,k')}$. When the contribution is positive, it pushes the weight $\boldsymbol{W}^V_{(j,k)}$ to **decrease**; when negative, it pushes the weight to **increase**. The total gradient is the sum of contributions over all pairs $(i, k')$ in the dataset. Analogous results hold for gradients w.r.t. $\boldsymbol{W}^M$.*

**Learning the Transitive Reachability.** The next-token loss $\ell^{(1)}(\mathcal{D})$ encourages the backbone matrix $\boldsymbol{W}^V$ to capture the observed reachability from the training data (Wang et al., 2024b). For a given pair $(i, k')$, when the transfer matrix entry $\boldsymbol{W}^T_{(b,k')}$ is large (indicating a confident $b \to k'$ transition), and the model predicts a lower probability for $k'$ than the ground truth along a path $i \to b \to k'$, the 2nd-step prediction loss $\ell^{(2)}(\mathcal{D})$ applies a negative gradient to $\boldsymbol{W}^V_{(j,b)}$ (i.e., increasing its weight), thereby strengthening the $b \rightsquigarrow j$ relation. Therefore, for any node $k$, when $\boldsymbol{W}^T_{(k,k')}$ captures the true adjacency relationship between $k$ and $k'$, the 2nd-step prediction enables the backbone to learn the *transitive reachability* from $k$ to $j$, based on the observed reachability from $k'$ to $j$ is learned by $\boldsymbol{W}^V_{(j,k')}$ by the 1st-step token prediction, and the adjacency $(k, k')$ is learned by the transfer layer $\boldsymbol{W}^T_{(k,k')}$. This shows that 2-token prediction could achieve higher-order reachability beyond the observed reachability of the next-token prediction.

**Learning the Adjacency.** While the next-token loss $\ell^{(1)}(\mathcal{D})$ directly encourages $\boldsymbol{W}^M$ to capture the adjacency relationship in the dataset (Wang et al., 2024b), the 2nd-step prediction loss $\ell^{(2)}(\mathcal{D})$ operates indirectly. For a given pair $(j, k')$, when $\boldsymbol{W}^T_{(k,k')}$ is large and the model underestimates the probability of the second-step node $k'$, the loss applies a negative gradient to $\boldsymbol{W}^M_{(i,k)}$, thereby strengthening the $i \to k$ connection. This suggests that *spurious adjacency* $(i, k)$ may be introduced into $\boldsymbol{W}^M$ when learning transitive reachability by the 2nd-step prediction, which is mechanically difficult to avoid due to the tight coupling between adjacency and reachability learning in the backbone. Our empirical validation later (Section 5) demonstrates that this risk is limited and the overall benefit of learning transitive reachability outweighs the risk of spurious adjacency.

**Lemma 1.** *At convergence, for each context $(i, j)$, the Softmax output probability distribution is uniquely determined and matches the empirical distribution, whereas the individual weight matrices are not uniquely determined.*

Proof follows from the derivations in Appendix C.2.

**Next-Node Prediction.** During the next-token prediction inference, the model samples the next node $k$ with high $\boldsymbol{W}^M_{(u_n,k)} + \boldsymbol{W}^V_{(u_2,k)}$, which favors nodes that are both neighbors of the current node (high $\boldsymbol{W}^M$) and reachable from the target (high $\boldsymbol{W}^V$), and correctly reflects the essence of path planning. Moreover, the transitive reachability learned from $\ell^{(2)}$ helps the model generate more accurate paths, leading to an improvement on its performance, especially for high-order test cases.

### 3.2 LEARNING MECHANISM OF MULTI-TOKEN PREDICTION

The total loss of MTP is defined as the sum of cross-entropy losses at each step: $\ell(\mathcal{D}) = \sum_{i=1}^{n} \ell^{(i)}(\mathcal{D})$, where each $\ell^{(i)}$ corresponds to an independent transfer layer. The transfer layer used for generating the outputs of the $n$-th step token is denoted as $\mathbf{W}^{T(n-1)}$.

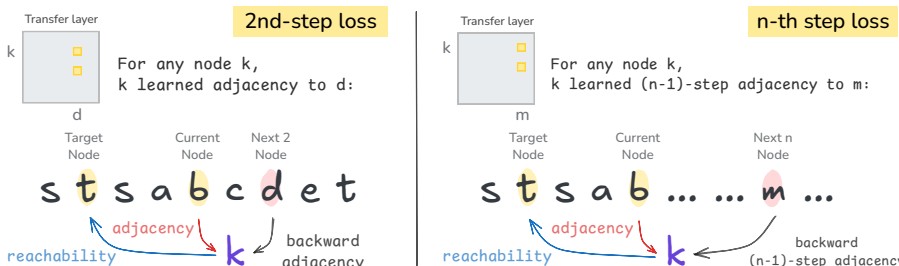

Figure 2: **Illustration of the learning mechanism under Multi-Token Prediction.** Left: 2nd-step loss; Right: $n$-th step loss. MTP learns transitive reachability and spurious adjacency.

Theorems 1 and 2 generalize naturally: the $n$-th transfer layer $\boldsymbol{W}^{T(n-1)}$ takes the next-step logits from the backbone model as input and outputs the $n$-th step logits, such that $\boldsymbol{W}^{T(n-1)}$ approximates the $(n-1)$-th power of the adjacency matrix. Under the influence of $\boldsymbol{W}^{T(n-1)}$, the model can capture the transitive reachability composed of the observed reachability from the $n$-th step node $m$ to target $t$ and the $(n-1)$-th power adjacency from some node $k$ to $m$ learned under $\boldsymbol{W}^{T(n-1)}$ (Figure 2). Meanwhile, it may learn some spurious adjacency from the current node $b$ to $k$.

## 4 ENHANCING TRANSFER LAYERS FOR MULTI-TOKEN PREDICTION

### 4.1 NEXT-TOKEN INJECTION

The transfer layer projects the backbone representation at the next-step position to predict future tokens. Its performance is constrained by the backbone output $\boldsymbol{h}_n$'s ability to predict the next node, where the deviation between the predicted and true next step introduces noise.

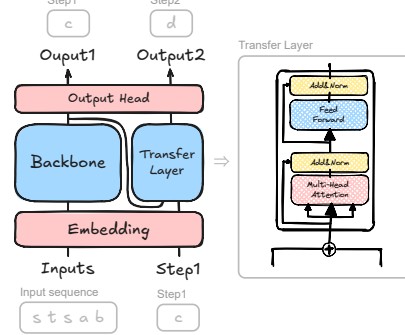

We propose **Next-Token Injection** (NTI), which augments the transfer input with information from the true next node to provide direct supervision. This is achieved by injecting the embedding vector of the true next token $u_{n+1}$ into the backbone's hidden state $\boldsymbol{h}_n$, which is then mapped to different positions by separate transfer layers, as follows:

Figure 3: Enhanced transfer layer architecture with NTI and Transformer-based Transfer Layer.

$$\widetilde{\boldsymbol{h}}_n = \boldsymbol{h}_n + k(\boldsymbol{W}_t)_{(:,u_{n+1})}, \ \text{logits}_2 = \boldsymbol{W}_o \widetilde{\boldsymbol{h}}_n \boldsymbol{W}^T, \quad (5)$$

where $k$ is a learnable scalar balancing the internal representation and external supervision.

NTI's residual shortcut reframes absolute prediction into a simple transformation, analogous to the shortcuts in ResNet (He et al., 2016), thus enabling gradients to bypass unstable backbone states and directly optimize the transfer layer. To analyze this from a gradient perspective, let $\widehat{\boldsymbol{p}}_{n+2} = \text{softmax}(\text{logits}_2)$ denote the predicted probability distribution, and let $\boldsymbol{e}_{n+2}$ be the one-hot vector for the true token $u_{n+2}$. The gradient of the loss with respect to the transfer matrix is then:

$$\frac{\partial \ell^{(2)}(\mathcal{D})}{\partial \boldsymbol{W}^T} = \left(\boldsymbol{W}_o \left(\boldsymbol{h}_n + k(\boldsymbol{W}_t)_{(:,u_{n+1})}\right)\right)^\top (\widehat{\boldsymbol{p}}_{n+2} - \boldsymbol{e}_{n+2}), \quad (6)$$

thus preserving the informativeness of supervision even when the predicted next-step hidden state is corrupted by noise, thereby enhancing stability and accuracy in structural modeling.

### 4.2 TRANSFORMER-BASED TRANSFER LAYER

The linear transfer layer independently maps each step of the sequence to the next step. We replace it with a **Transformer-based Transfer Layer**, whose input is the hidden representations produced by the backbone at each step: $\boldsymbol{H} = [\boldsymbol{h}_1, \boldsymbol{h}_2, \ldots, \boldsymbol{h}_N] \in \mathbb{R}^{N \times d}$. The Transformer captures dependencies across sequence positions through self-attention, allowing each step's representation to dynamically integrate information from the preceding context.

Let the output of the transfer layer be $\widetilde{\boldsymbol{H}} = \mathcal{T}(\boldsymbol{H})$, and denote the loss at step $n$ as $\ell_n^{(2)}(\mathcal{D})$. The gradient with respect to the backbone hidden states is then:

$$\frac{\partial \ell_n^{(2)}(\mathcal{D})}{\partial \boldsymbol{H}} = \frac{\partial \ell_n^{(2)}(\mathcal{D})}{\partial \widetilde{\boldsymbol{H}}} \cdot \frac{\partial \mathcal{T}(\boldsymbol{H})}{\partial \boldsymbol{H}}. \tag{7}$$

We denote the self-attention weight matrix at step $n$ as $\boldsymbol{A}_n \in \mathbb{R}^{N \times N}$, which represents the dependencies of $\boldsymbol{h}_n$ on all steps in the sequence, then the gradient for each step $i$ can be expressed as

$$\frac{\partial \ell_n^{(2)}(\mathcal{D})}{\partial \boldsymbol{h}_i} = \sum_{j=i}^{N} \boldsymbol{A}_n(i,j) \frac{\partial \ell_n^{(2)}(\mathcal{D})}{\partial \widetilde{\boldsymbol{h}}_j}, \quad i = 1, \ldots, N. \tag{8}$$

Consequently, the gradient for each step aggregates errors from subsequent steps, enabling the backbone model to fully leverage the sequence context during training, thereby improving overall performance and training stability.

The overall architecture of the enhanced transfer layer, combining Next-Token Injection and Transformer-based Transfer Layer, is illustrated in Figure 3.

## 5 EMPIRICAL EVALUATION ON GRAPH PLANNING

### 5.1 OVERALL ACCURACY OF DIFFERENT MODELS ON PATH PLANNING

We evaluate model performance on randomly generated directed acyclic graphs (DAGs) by measuring prediction accuracy on test paths. To analyze performance under varying planning difficulties, test paths are categorized into degree-0/1/2/3 classes according to their reachability in the observation graph $\mathcal{G}_{\text{obs}}$, as defined in Section 2. The code is available at https://anonymous.4open.science/r/mtp-for-alpine. Additional experiments on directed graphs with cycles are provided in Appendix D.

**Experimental Setup:** For each trial, we generate a random DAG with $n = 100$ nodes, where each potential edge $(i, j)$ for $i < j$ is included independently with probability $p = 0.1$. For every reachable source–target pair $(s, t)$, we randomly sample $m = 20$ valid paths. To increase the number of test paths, 10% of $(s, t)$ pairs are used for training and the remaining 90% for testing, while all one-hop edges $(s, t) \in \mathcal{E}$ are always included in the training set as direct paths "$s\ t\ s\ t \backslash n$". All models use 120-dimensional embeddings and adopt a 1-layer, 1-head Transformer as the backbone. "NTI" denotes models with Next-Token Injection (Section 4). The Transformer-based Transfer Layer uses the same hidden size as the backbone and varies in depth (1, 3, or 6 layers).

**Metrics:** We evaluate our models using three metrics. **Graph-level accuracy** is computed by first averaging the path-level accuracy within each graph, and then averaging across all graphs. **Standard error** is calculated by dividing the standard deviation of graph-level accuracies by the square root of the total number of graphs. **Path-level accuracy** is the average accuracy over all test paths.

**Results:** As shown in Table 1, the MTP models achieve notable improvements over the 1-Token Prediction (i.e., Next-Token Prediction) baseline on degree-2/3 test paths. Compared to the MTP baseline, incorporating NTI and Transformer-based Transfer Layers consistently boosts performance across all degrees. The significant gains in accuracy for degree-2/3 tests in particular demonstrate the effectiveness of MTP in handling transitive reachability. On degree-2 tests, 3-Token models often underperform 2-Token models, as 2-Token Predictions already capture the required reachability; additional tokens provide limited benefit and may introduce minor training noise, slightly reducing performance. These results align with our theoretical analysis.

**Effect of Backbone Architecture:** We investigate the impact of backbone model complexity, including Transformer depth and number of attention heads. As shown in Figure 4, increasing backbone model complexity yields only slight changes (also shown in Wang et al. (2024b)). In contrast, the configuration combining 2-Token, NTI, and a Transformer-based Transfer Layer consistently yields stable accuracy gains.

Table 1: **Path prediction accuracy (%) on degree-0/1/2/3 paths in 100-node DAGs.** Metrics include graph-level accuracy (with $\pm$ standard error) and path-level accuracy. Results for degree-0/1 are averaged over 50 graphs; degree-2/3 over 200 graphs.

| MODEL | DEGREE-0 Graph $\pm$ / Path | DEGREE-1 Graph $\pm$ / Path | DEGREE-2 Graph $\pm$ / Path | DEGREE-3 Graph $\pm$ / Path | OVERALL Path |
|---|---|---|---|---|---|
| 1-Token (baseline) (Wang et al., 2024b) | 92.58$\pm$0.18 / 92.56 | 86.57$\pm$0.24 / 86.60 | 63.80$\pm$0.65 / 64.34 | 30.76$\pm$1.30 / 33.25 | 89.31 |
| 2-Token (Meta's) (Gloeckle et al., 2024) | 92.03$\pm$0.19 / 92.00 | 85.54$\pm$0.27 / 85.60 | 66.37$\pm$0.56 / 66.78 | 36.09$\pm$1.33 / 35.55 | 88.65 |
| 2-Token (DeepSeek's) (Liu et al., 2024) | 93.71$\pm$0.18 / 93.62 | 88.82$\pm$0.22 / 88.79 | 67.78$\pm$0.58 / 68.48 | 31.85$\pm$1.34 / 34.36 | 90.90 |
| 2-Token + NTI (linear transfer layer) | 94.09$\pm$0.24 / 94.14 | 90.00$\pm$0.29 / 90.02 | 69.51$\pm$0.60 / 69.56 | 39.25$\pm$1.28 / 42.08 | 91.74 |
| 2-Token + 1-layer Transformer | 93.87$\pm$0.15 / 93.83 | 87.84$\pm$0.25 / 87.86 | 68.78$\pm$0.55 / 69.51 | 37.11$\pm$1.25 / 39.26 | 90.68 |
| 2-Token + NTI + 1-layer | **96.43$\pm$0.12 / 96.43** | 88.92$\pm$0.21 / 88.94 | 71.32$\pm$0.51 / 71.52 | 43.49$\pm$1.17 / 44.37 | **92.65** |
| 2-Token + NTI + 3-layer | 93.76$\pm$0.16 / 93.70 | 89.47$\pm$0.24 / 89.39 | 71.41$\pm$0.55 / 71.60 | 43.43$\pm$1.34 / 44.26 | 91.29 |
| 2-Token + NTI + 6-layer | 94.56$\pm$0.17 / 94.50 | **90.09$\pm$0.23 / 90.10** | **72.56$\pm$0.47 / 72.97** | **43.81$\pm$1.23 / 45.70** | 92.07 |
| 3-Token (Meta's) (Gloeckle et al., 2024) | 90.76$\pm$0.22 / 90.72 | 83.37$\pm$0.29 / 83.34 | 62.17$\pm$0.58 / 62.24 | 34.63$\pm$1.22 / 37.23 | 86.90 |
| 3-Token (DeepSeek's) (Liu et al., 2024) | **94.45$\pm$0.16 / 94.37** | 89.42$\pm$0.25 / 89.43 | 66.21$\pm$0.60 / 66.42 | 30.22$\pm$1.29 / 32.27 | **91.54** |
| 3-Token + NTI (linear transfer layer) | 92.19$\pm$0.15 / 92.17 | 87.37$\pm$0.20 / 87.39 | 63.38$\pm$0.56 / 64.06 | 42.96$\pm$1.36 / 46.28 | 89.44 |
| 3-Token + 1-layer Transformer | 92.22$\pm$0.15 / 92.16 | 84.11$\pm$0.22 / 84.15 | 66.79$\pm$0.47 / 67.22 | 40.67$\pm$1.24 / 40.43 | 88.17 |
| 3-Token + NTI + 1-layer | 92.35$\pm$0.16 / 92.31 | 85.34$\pm$0.24 / 85.37 | 69.61$\pm$0.49 / 70.10 | 44.82$\pm$1.31 / 46.10 | 88.84 |
| 3-Token + NTI + 3-layer | 93.29$\pm$0.13 / 93.25 | 89.54$\pm$0.16 / 89.52 | 71.97$\pm$0.47 / 72.39 | **45.38$\pm$1.18 / 47.25** | 91.12 |
| 3-Token + NTI + 6-layer | 93.55$\pm$0.14 / 93.52 | **89.67$\pm$0.18 / 89.66** | **72.82$\pm$0.49 / 73.09** | 45.18$\pm$1.24 / 46.99 | 91.34 |

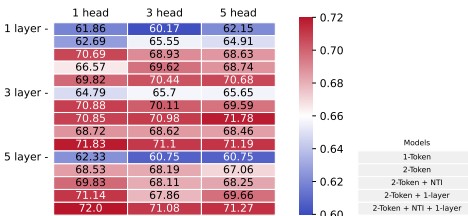

Figure 4: Degree-2 path graph-level accuracy (%) for Transformer depth and number of heads, averaged over 10 fixed graphs.

Table 2: Degree-2 path graph-level accuracy (with $\pm$ standard error) for node counts and embedding sizes, averaged over 100 graphs.

| MODEL | 100-NODE 120-DIM | 200-NODE 120-DIM | 300-NODE 120-DIM | 300-NODE 320-DIM |
|---|---|---|---|---|
| 1-Token | 63.00$\pm$0.89 | 60.86$\pm$0.82 | **56.16$\pm$0.64** | 55.92$\pm$0.64 |
| 2-Token | 65.99$\pm$0.88 | 62.43$\pm$0.60 | 50.13$\pm$0.69 | 57.47$\pm$0.80 |
| 2-Token + NTI | 69.71$\pm$0.85 | 63.84$\pm$0.81 | 53.79$\pm$0.60 | 65.32$\pm$0.71 |
| 2-Token + 1-layer | 69.02$\pm$0.69 | 61.30$\pm$0.66 | 46.62$\pm$0.52 | 60.51$\pm$0.68 |
| 2-Token + NTI + 1-layer | **71.55$\pm$0.73** | **65.10$\pm$0.62** | 49.79$\pm$0.54 | **65.14$\pm$0.76** |

**Effect of Graph Size:** We further evaluate scalability on graphs with 200 and 300 nodes, using a fixed embedding size of 120 (shown in Table 2). Results show that 2-Token performance degrades with increasing graph size. On 300-node graphs, increasing the embedding dimension to 320 enables the 2-Token model to outperform the 1-Token baseline. When further increasing the graph size and allocating a matching embedding dimension, the experimental results exhibit the same behavior. It is likely that when the number of nodes exceeds the embedding capacity, supervision signals from different tokens may conflict, limiting the model's ability to encode structural patterns.

## 5.2 WEIGHT ANALYSIS OF THE TRAINED MODEL

In a 100-node path-planning task, we project the linear transfer layer $\boldsymbol{W}^T$ onto the node representations via $\boldsymbol{W}_t \boldsymbol{W}^T \boldsymbol{W}_o$ to evaluate its ability to learn transition relations, where $\boldsymbol{W}_t$ and $\boldsymbol{W}_o$ denote the input embedding and output projection matrices, respectively. Figure 5 illustrates the visual projection and Table 3 reports the average weights of the projection under true adjacency and non-adjacency entries. The results clearly show that the transfer layer $\boldsymbol{W}^T$ is learning the true adjacency and with NTI the learning effect is much better.

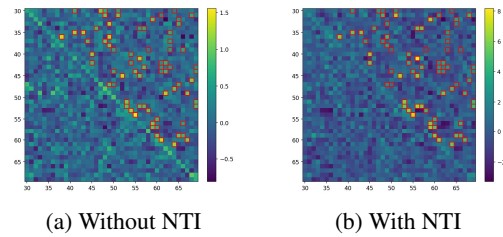

(a) Without NTI  (b) With NTI

Figure 5: **Visualization of the projected matrix on the 100-node graph without and with NTI.** Red boxes indicate ground-truth adjacency, showing only the central submatrix.

In the simplified 100-node model, with the transfer layer $\boldsymbol{W}^T$ fixed to the ground-truth adjacency matrix, we visualize the first 20 nodes of the learned weight matrices $\boldsymbol{W}^M$ and $\boldsymbol{W}^V$, as shown in Figures 6a and 6b. In $\boldsymbol{W}^M$, this region fully corresponds to relations that are theoretically learnable under $\ell^{(2)}(\mathcal{D})$, with some weights slightly amplified, though still far less pronounced than those reinforced by $\ell^{(1)}(\mathcal{D})$. In $\boldsymbol{W}^V$, a sub-

set of relations is captured by $\ell^{(2)}(\mathcal{D})$, with their weights clearly standing out above the background (Table 4).

Table 3: Average weights of adjacency and non-adjacency entries in the projected matrix and their gap.

| SETTING | ADJ. AVG | NON-ADJ. AVG | GAP |
|---|---|---|---|
| Without NTI | 0.82 | -0.01 | 0.83 |
| With NTI | 4.01 | -0.05 | 4.06 |

The 2nd-step prediction encourages $\boldsymbol{W}^M$ to assign small positive weights to certain spurious adjacency relations, introducing mild noise. In contrast, $\boldsymbol{W}^V$ significantly strengthens correct but unobserved reachability relations, with weights clearly standing out from the background nonreachable entities. These enhanced reachability signals are concentrated near the diagonal in key positions, making them substantially

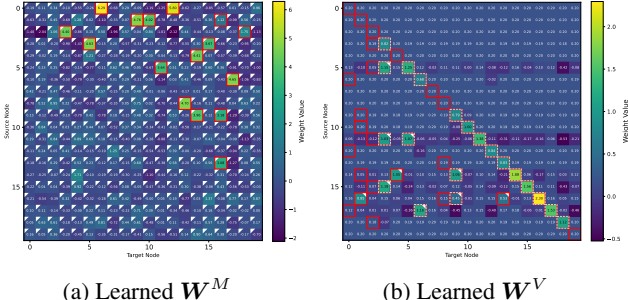

(a) Learned $\boldsymbol{W}^M$      (b) Learned $\boldsymbol{W}^V$

Figure 6: **Weight analysis on the first 20 nodes of the 100-node graph using a simplified model with fixed transfer layer $\boldsymbol{W}^T$.** Red boxes indicate true adjacency and reachability, light dashed boxes show observed reachability, and white triangles mark theoretically learnable entries under $\ell^{(2)}(\mathcal{D})$.

more influential during the next-step prediction process. This indicates that the 2nd-step prediction indeed enables the backbone model to learn transitive reachability beyond those learned by the next-token prediction. Further visualizations and analyses of weights for the general 2-Token Prediction can be found in Appendix E.

## 5.3 EVALUATION ON THE BLOCKSWORLD PLANNING TASK

To further assess the practicality and generalization of our methods, we evaluate their performance on the classical Blocksworld planning benchmark. This task provides a structured, fixed-graph environment to test the models' ability in finding valid action sequences. Full details of the experimental setup, including the graph representation and dataset construction methodology, are provided in Appendix F.

Table 4: Average weights of different entry types in $\boldsymbol{W}^M$ and $\boldsymbol{W}^V$ in the 100-node graph with a fixed transfer layer $\boldsymbol{W}^T$ in the simplified model.

| MATRIX | TYPE | VALUE |
|---|---|---|
| $\boldsymbol{W}^M$ | True adjacency | 1.90 |
| | Theoretically learnable | 0.16 |
| | Other entries | -0.01 |
| $\boldsymbol{W}^V$ | Observed reachability | 0.63 |
| | Theoretically learnable | 0.34 |
| | Other entries | -0.01 |

**Results:** As shown in Table 5, our proposed enhancements to Multi-Token Prediction consistently outperform the 1-token baseline and standard MTP architectures across various training data sizes. Notably, the combination of NTI and a multi-layer Transformer transfer layer (e.g., 2-Token + NTI + 6-layer) achieves the highest accuracy, demonstrating the effectiveness of our approach in a complex, classical planning environment.

## 6 DISCUSSION AND FUTURE WORK

In summary, the paper provides an in-depth analysis of how Multi-Token Prediction enables the autoregressive Transformer to learn transitive relations in graph path planning. Based on this, we propose two enhancement strategies: Next-Token Injection (NTI) and a Transformer-based Transfer Layer. Experiments on synthetic graphs and the Blocksworld planning task demonstrate that these methods significantly improve the model's accuracy and stability in transitive planning tasks.

**Challenges in Controllable Learning:** Using 2-Token Prediction enables explicit learning of transitive reachability relations, yet the newly injected gradients still induce only minor changes to the backbone Transformer's weights. While Multi-Token Prediction can supplement unobserved

Table 5: Path prediction accuracy (%) on Blocksworld under varying training set sizes.

| Train Size (Number of Paths per Length) | 100 | 200 | 300 | 400 | 500 |
|---|---|---|---|---|---|
| 1-Token (baseline) | 45.62 | 62.66 | 70.21 | 74.86 | 77.94 |
| 2-Token (Meta's) (Gloeckle et al., 2024) | 42.42 | 60.40 | 68.19 | 72.31 | 76.27 |
| 2-Token (DeepSeek's) (Liu et al., 2024) | 51.32 | 66.60 | 74.01 | 78.41 | 80.23 |
| 2-Token + NTI (linear transfer layer) | 44.91 | 62.46 | 71.36 | 74.85 | 80.03 |
| 2-Token + 1-layer Transformer | 51.25 | 66.51 | 75.84 | 77.11 | 79.32 |
| 2-Token + NTI + 1-layer | **52.51** | 67.37 | **75.92** | **79.92** | 81.55 |
| 2-Token + NTI + 3-layer | 52.01 | 66.73 | 74.86 | 79.56 | 83.30 |
| 2-Token + NTI + 6-layer | 52.84 | **68.57** | 73.74 | 78.97 | **85.70** |
| 3-Token (Meta's) (Gloeckle et al., 2024) | 40.99 | 56.41 | 64.67 | 69.38 | 76.43 |
| 3-Token (DeepSeek's) (Liu et al., 2024) | 50.13 | 65.30 | 71.52 | 76.87 | 80.87 |
| 3-Token + NTI (linear transfer layer) | 42.91 | 62.46 | 68.47 | 73.82 | 78.29 |
| 3-Token + 1-layer Transformer | 49.98 | 64.95 | 72.63 | 76.83 | 78.37 |
| 3-Token + NTI + 1-layer | **50.47** | 67.13 | **72.88** | 76.45 | 81.48 |
| 3-Token + NTI + 3-layer | 49.84 | 66.57 | 71.70 | **77.47** | 80.40 |
| 3-Token + NTI + 6-layer | 50.10 | 67.11 | 72.69 | 77.27 | **82.18** |

structural information, controlling the suppression of spurious adjacency relations while strengthening transitive reachability remains an open problem. Future work may explore combining MTP with additional mechanisms, such as reward signals from reinforcement learning, chain-of-thought reasoning, or backtracking search, to provide more effective mechanisms to shape the learning dynamics.

**Implications for Multi-Token Prediction:**   In the ALPINE task, attention is restricted to the target node, so the backbone Transformer primarily focuses on information regarding the current and target nodes. Multi-Token Prediction mainly supplements adjacency and reachability relations. In more general tasks, attention is distributed across the sequence. Here, the analysis can be extended: the transfer layer performs multi-step structured transformations on hidden states, allowing the loss to propagate back and inject longer-range, structurally informative gradients. Due to the non-uniqueness of learned parameters, the effectiveness and boundaries of this mechanism in complex tasks remain to be explored.

**Practical Extensions to Real-World Tasks:**   Although our analysis is conducted on synthetic graphs and structured planning domains, the insights naturally extend to more realistic settings. Real-world environments often exhibit continuous, noisy, or semantically ambiguous states, posing a challenge in abstracting them into discrete structures suitable for multi-step prediction. Future work may explore applying Multi-Token Prediction to domains such as visual navigation, robotic manipulation, or text-based planning by incorporating appropriate state discretization or structural extraction methods. This direction would help assess the generality and robustness of the proposed mechanisms in more complex, real-world scenarios.

These findings not only demonstrate the effectiveness of Multi-Token Prediction in structured planning tasks but also highlight its potential to facilitate complex reasoning and structured understanding in large language models. Future work will investigate applying these mechanisms to more complex and realistic domains. We hope this study encourages the community to combine theoretical and empirical approaches when studying LLMs, ultimately advancing their planning capabilities and structural comprehension.

ETHICS STATEMENT

All authors have adhered to the ICLR Code of Ethics. Our work is theoretical and empirical, using only synthetic graph data and the Blocksworld benchmark, with no human subjects or sensitive information, and poses no societal risk. We believe it supports responsible development of capable and interpretable AI systems.

REPRODUCIBILITY STATEMENT

We are committed to ensuring reproducibility. For experiments, an anonymous link is provided in Section 5.1 with the full source code, data generation scripts, and model architectures and hyperparameters. For theory, all derivations and proofs are provided in the appendices to allow verification of our claims.

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

# A  RELATED WORK

## A.1  LLMS FOR STRUCTURED PLANNING AND REASONING

Recent studies have examined the capacity of large language models (LLMs) to perform structured planning and reasoning. In planning domains, benchmarks such as CogEval (Momennejad et al., 2023) and Blocksworld (Valmeekam et al., 2023) highlight significant challenges: while humans achieve over 70% success in Blocksworld, GPT-3 attains only 5%, suggesting that LLMs struggle to capture underlying task structures. Nevertheless, LLMs exhibit promising behaviors in multi-step decision-making for autonomous agents (Wang et al., 2024a), which can often be abstracted as path-finding over graphs. For example, HuggingGPT (Shen et al., 2023) coordinates external APIs through dependency relations, naturally forming a graph-based planning problem. Our work adopts this abstraction but focuses on consistency and interpretability in multi-step prediction.

Beyond planning, researchers have explored the graph reasoning abilities of LLMs. Frameworks such as GPT4Graph (Guo et al., 2023) and NLGraph (Wang et al., 2023) show that while LLMs can process graph-structured inputs, performance remains fragile and sensitive to spurious correlations, with GPT-4 reaching only around 50% accuracy on shortest-path tasks. To improve reasoning, recent efforts augment LLMs with external modules such as GNN encoders (Chai et al., 2023; Tang et al., 2024) or explicitly train them to imitate classical algorithms like BFS and DFS (Luo et al., 2024). Other work focuses on extracting structured task knowledge, e.g., distilling temporal and causal relations into compact representations (Yuan et al., 2023). While these approaches improve empirical performance, they provide limited insights into why LLMs fail on more complex planning scenarios.

A complementary line of research examines the algorithmic foundations of LLM reasoning. Transformers have been shown to belong to the $\mathrm{TC}^0$ complexity class (Merrill & Sabharwal, 2023), but techniques like chain-of-thought prompting (Feng et al., 2023) and Tree of Thoughts search (Yao et al., 2023) allow them to simulate more complex procedures sequentially. Parsel (Zelikman et al., 2023), for instance, decomposes reasoning into structured subroutines, closely related to the multi-step prediction framework considered here. However, these studies largely overlook how the autoregressive training paradigm itself may impose fundamental limitations on consistent multi-step planning—a gap that our theoretical analysis seeks to address.

## A.2  MULTI-TOKEN PREDICTION (MTP)

Traditional language models are trained with autoregressive next-token prediction (NTP), which requires generating tokens sequentially during inference, leading to low generation efficiency (Allen-Zhu & Li, 2023). To accelerate inference, **Multi-Token Prediction (MTP)** has been proposed as an alternative paradigm. Unlike traditional approaches that predict a single token at each step, MTP predicts multiple future tokens in parallel, typically through several output heads attached to a shared Transformer backbone (Gloeckle et al., 2024; Cai et al., 2024).

The main advantage of MTP lies in its ability to improve inference efficiency. By generating multiple candidate tokens in parallel and adopting *self-speculative decoding*, approaches such as Medusa (Cai et al., 2024) and Hydra (Ankner et al., 2024) achieve up to $3\times$ speedup in low-batch scenarios. Moreover, the multi-step prediction mechanism of MTP can partially alleviate exposure bias, leading to modest performance gains. In tasks such as code generation and speech modeling, MTP-trained models not only decode faster but also exhibit better multi-step consistency compared to autoregressive models (Gloeckle et al., 2024; Wang et al., 2025).

Unlike prior studies that mainly focus on inference acceleration, this work leverages the ALPINE framework to investigate how the MTP mechanism affects the learning behavior of the backbone model. Through theoretical analysis, we explore how multi-token supervision alters representation formation and optimization dynamics during training, thereby revealing its potential advantages in model learning.

### A.3 AUGMENTING LEARNING WITH FUTURE INFORMATION

Increasing evidence indicates that the short-sightedness of Next-Token Prediction (NTP) is a key limitation for complex reasoning and planning. By focusing on local statistical patterns, models trained with NTP often fail to acquire a global understanding of long-range structure and causal relations (Bachmann & Nagarajan, 2024). To address this, several paradigms inject information beyond the immediate next token into the training signal.

These approaches vary in form: some directly predict raw tokens at future positions to introduce sparse long-range signals, such as Next-X Prediction (Ren et al., 2025), L-MTP (Liu et al., 2025), and Trelawney (Thankaraj et al., 2025); others learn more abstract future representations, e.g., by predicting compressed summaries of subsequent text (Mahajan et al., 2025) or by leveraging multi-step future information encoded in hidden states (Pal et al., 2023). Incorporating such future information enables models to integrate broader context during generation and reasoning, thereby improving their capacity for complex reasoning tasks.

Prior studies primarily analyze the performance gains of these methods from an empirical perspective, whereas our work takes a complementary view by examining the *learning dynamics*. Through gradient-based analysis, we investigate how future-token signals influence representation formation and optimization, providing a deeper understanding of how MTP enhances model capabilities by injecting future tokens.

## B SIMPLIFIED TRANSFORMER SETUP

To theoretically analyze how the 2-Token Prediction objective helps the model learn structural information, we follow the analysis framework of (Wang et al., 2024b), starting from a standard Transformer model and gradually simplifying it into an analytically tractable form. We begin by reviewing the standard Transformer computations and then describe the simplifications introduced.

In the original model, the input tokens $x_1, x_2, \ldots, x_N$ are mapped into embedding vectors $\boldsymbol{X} \in \mathbb{R}^{N \times d}$, which are then processed through a stack of Transformer blocks. Each block consists of a multi-head self-attention (MHA) module and a feed-forward network (FFN). The attention mechanism is defined as:

$$\text{Attention}(\boldsymbol{Q}, \boldsymbol{K}, \boldsymbol{V}) = \text{softmax}\left(\frac{\boldsymbol{Q}\boldsymbol{K}^\top}{\sqrt{d_k}}\right)\boldsymbol{V}, \tag{9}$$

where $\boldsymbol{Q}, \boldsymbol{K}, \boldsymbol{V} \in \mathbb{R}^{N \times d_k}$ are the query, key, and value matrices respectively, $d_k = d/H$ is the per-head embedding dimension, and $H$ is the number of attention heads. For input $\boldsymbol{X} \in \mathbb{R}^{N \times d}$, the $i$-th head computes: $\boldsymbol{Q}_i = \boldsymbol{X}\boldsymbol{W}_i^Q, \quad \boldsymbol{K}_i = \boldsymbol{X}\boldsymbol{W}_i^K, \quad \boldsymbol{V}_i = \boldsymbol{X}\boldsymbol{W}_i^V$, with learnable parameters $\boldsymbol{W}_i^Q, \boldsymbol{W}_i^K, \boldsymbol{W}_i^V \in \mathbb{R}^{d \times d_k}$. The multi-head attention output is the concatenation of all heads: $\text{MHA}(\boldsymbol{X}) = \text{Concat}_{i=1}^H\left(\text{Attention}(\boldsymbol{Q}_i, \boldsymbol{K}_i, \boldsymbol{V}_i)\right)$.

The feed-forward network is a two-layer MLP:

$$\text{FFN}(\boldsymbol{X}) = \max(\boldsymbol{0}, \boldsymbol{X}\boldsymbol{W}_1 + \boldsymbol{1}\boldsymbol{b}_1^\top)\boldsymbol{W}_2 + \boldsymbol{1}\boldsymbol{b}_2^\top, \tag{10}$$

where $\boldsymbol{W}_1 \in \mathbb{R}^{d \times 4d}$, $\boldsymbol{W}_2 \in \mathbb{R}^{4d \times d}$, $\boldsymbol{b}_1 \in \mathbb{R}^{4d}$, $\boldsymbol{b}_2 \in \mathbb{R}^d$, $\boldsymbol{1} \in \mathbb{R}^{N \times 1}$ is the all-one vector for broadcasting biases, and $\boldsymbol{0} \in \mathbb{R}^{N \times 4d}$ is the zero matrix used in the ReLU activation (Glorot et al., 2011).

Each Transformer block applies residual connections and layer normalizations:

$$\text{Transformer}(\boldsymbol{X}) = \text{FFN}(\text{LN}_2(\text{MHA}(\text{LN}_1(\boldsymbol{X})) + \boldsymbol{X})) + \text{MHA}(\text{LN}_1(\boldsymbol{X})) + \boldsymbol{X}. \tag{11}$$

To enable tractable visualization and analysis, we structurally simplify the architecture. First, we retain only a single-layer, single-head self-attention module. The attention matrix $\text{softmax}\left(\frac{\boldsymbol{Q}\boldsymbol{K}^\top}{\sqrt{d_k}}\right)$ is manually set to a one-hot matrix in which the second column is filled with ones and all other entries are zero, i.e.,

$$\text{softmax}\left(\frac{\boldsymbol{Q}\boldsymbol{K}^\top}{\sqrt{d_k}}\right) = \begin{bmatrix} 0 & 1 & 0 & \cdots & 0 \\ 0 & 1 & 0 & \cdots & 0 \\ \vdots & \vdots & \vdots & \ddots & \vdots \\ 0 & 1 & 0 & \cdots & 0 \end{bmatrix} \in \mathbb{R}^{n \times n}. \tag{12}$$

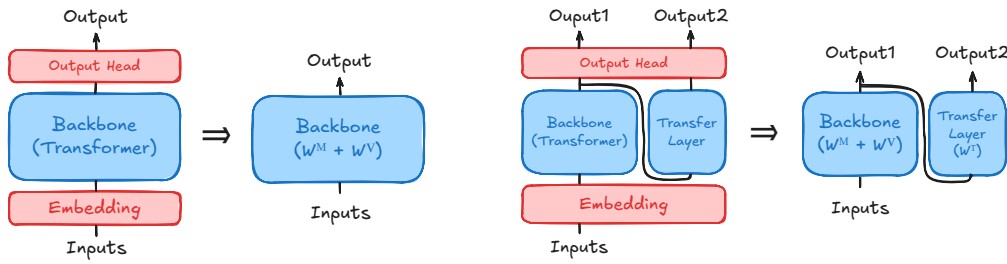

(a) Simplified 1-Token Prediction model      (b) Simplified 2-Token Prediction model

Figure 7: Illustration for the simplified model architectures.

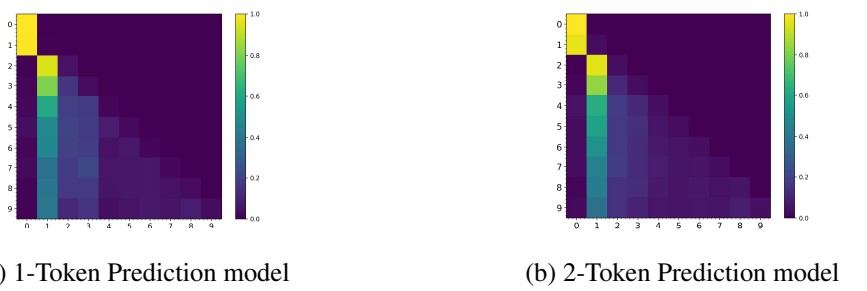

(a) 1-Token Prediction model      (b) 2-Token Prediction model

Figure 8: Visualization of attention matrices for (a) 1-Token Prediction and (b) 2-Token Prediction Transformer models.

This setting simulates the attention matrix learned by the model after training on the task, where each token attends exclusively to the target node (i.e., the second token in the sequence). In the actual path-planning task with 100 nodes, the relevant results are shown in Figure 8. These results are obtained by analyzing the attention mechanism of a single-layer single-head Transformer model, presenting the averaged attention matrix computed over the test dataset. Each row $n$ of the matrix corresponds to the attention distribution vector when predicting next token.

We remove all positional encodings by setting $\boldsymbol{W}_p = 0$, eliminate all layer normalization operations, and replace the two-layer FFN with a single linear transformation:

$$\text{FFN}(\boldsymbol{X}) = \boldsymbol{X}\boldsymbol{W}^M, \tag{13}$$

the forward propagation becomes an additive form. The resulting Transformer block is:

$$\text{Transformer}(\boldsymbol{X}) = \text{FFN}(\boldsymbol{X}) + \text{MHA}(\boldsymbol{X}). \tag{14}$$

To further reduce complexity, we set both the token embedding matrix $\boldsymbol{W}_t$ and the output projection matrix $\boldsymbol{W}_o$ to identity matrices, and assume that the embedding dimension equals the vocabulary size, i.e., $d = M$. This allows direct interpretation of logits in the vocabulary space.

Finally, in the 2-Token Prediction setting, we introduce a transfer matrix $\boldsymbol{W}^T \in \mathbb{R}^{M \times M}$ after the logits layer. This transfer layer is used to map the predicted next-step logits to the logits for the following step. The two resulting model variants are shown in Figure 7.

## C  DERIVATION AND CONVERGENCE ANALYSIS OF GRADIENTS

### C.1  DERIVATIONS AND PROOFS

Let $N_{i,j,k'}$ denote the number of times in $\mathcal{D}$ that the following conditions are satisfied: (a) the current node is $i$; (b) the attention target is $j$; and (c) the token two steps ahead is $k'$. Let $N_{i,j} = \sum_{k'} N_{i,j,k'}$ denote the total count of such $(i, j)$ pairs.

This leads to the following theorem:

**Theorem 1.** *For any pair $(i,j)$ in the dataset $\mathcal{D}$ with $N_{i,j} > 0$, let $P_{i,j}^{\mathrm{data}}(k') = \frac{N_{i,j,k'}}{N_{i,j}}$ denote the empirical probability of the second-next node $k'$. The contribution of this pair to the gradient $\frac{\partial \ell^{(2)}(\mathcal{D})}{\partial \boldsymbol{W}_{(d,k')}^T}$ is proportional to the prediction error $(\widehat{P}_{i,j}(k') - P_{i,j}^{\mathrm{data}}(k'))$, the sample count $N_{i,j}$, and the combined weight $(\boldsymbol{W}_{(i,d)}^M + \boldsymbol{W}_{(j,d)}^V)$. When the contribution is positive, it pushes the weight $\boldsymbol{W}_{(d,k')}^T$ to **decrease**; when negative, it pushes the weight to **increase**. The total gradient is the sum of contributions over all pairs $(i,j)$ in the dataset.*

**Theorem 2.**

*For any pair $(i,j)$ in dataset $\mathcal{D}$ with $N_{i,j} > 0$, the contribution of each (current node $i$, second-step node $k'$) pair to the gradient $\frac{\partial \ell^{(2)}(\mathcal{D})}{\partial \boldsymbol{W}_{(j,k)}^V}$ is proportional to the prediction error $(\widehat{P}_{i,j}(k') - P_{i,j}^{\mathrm{data}}(k'))$, the sample count $N_{i,j}$, and the weight $\boldsymbol{W}_{(k,k')}^T$. When the contribution is positive, it pushes the weight $\boldsymbol{W}_{(j,k)}^V$ to **decrease**; when negative, it pushes the weight to **increase**. The total gradient is the sum of contributions over all pairs $(i,k')$ in the dataset.*

*For any pair $(i,j)$ in dataset $\mathcal{D}$ with $N_{i,j} > 0$, the contribution of each (target node $j$, second-step node $k'$) pair to the gradient $\frac{\partial \ell^{(2)}(\mathcal{D})}{\partial \boldsymbol{W}_{(i,k)}^M}$ is proportional to the prediction error $(\widehat{P}_{i,j}(k') - P_{i,j}^{\mathrm{data}}(k'))$, the sample count $N_{i,j}$, and the weight $\boldsymbol{W}_{(k,k')}^T$. When the contribution is positive, it pushes the weight $\boldsymbol{W}_{(i,k)}^M$ to **decrease**; when negative, it pushes the weight to **increase**. The total gradient is the sum of contributions over all pairs $(j,k')$ in the dataset.*

*Proof.*

According to the definition of cross-entropy loss and the predicted weight vectors in our simplified model, the total cross-entropy loss (involving matrices $\boldsymbol{W}^M$, $\boldsymbol{W}^V$, and $\boldsymbol{W}^T$) is given by

$$\ell^{(2)}(\mathcal{D}) = -\sum_{u \in \mathcal{D}} \sum_{n=1}^{N-2} \sum_k \boldsymbol{U}_{(n+2,k)} \log \frac{\exp\left((\boldsymbol{W}^M \boldsymbol{W}^T)_{(u_n,k)} + (\boldsymbol{W}^V \boldsymbol{W}^T)_{(u_2,k)}\right)}{\sum_\ell \exp\left((\boldsymbol{W}^M \boldsymbol{W}^T)_{(u_n,\ell)} + (\boldsymbol{W}^V \boldsymbol{W}^T)_{(u_2,\ell)}\right)}$$

Let $\quad \boldsymbol{A}_{(i,k)} = \left(\boldsymbol{W}^M \boldsymbol{W}^T\right)_{(i,k)}, \quad \boldsymbol{B}_{(j,k)} = \left(\boldsymbol{W}^V \boldsymbol{W}^T\right)_{(j,k)}$

$$\ell^{(2)}(\mathcal{D}) = -\sum_{u \in \mathcal{D}} \sum_{n=1}^{N-2} \sum_k \boldsymbol{U}_{(n+2,k)} \sum_{i,j} \mathbb{I}[u_n = i, u_2 = j] \log \frac{\exp(\boldsymbol{A}_{(i,k)} + \boldsymbol{B}_{(j,k)})}{\sum_\ell \exp(\boldsymbol{A}_{(i,\ell)} + \boldsymbol{B}_{(j,\ell)})}$$

$$= -\sum_{i,j,k'} N_{i,j,k'} \log \frac{\exp(\boldsymbol{A}_{(i,k')} + \boldsymbol{B}_{(j,k')})}{\sum_\ell \exp(\boldsymbol{A}_{(i,\ell)} + \boldsymbol{B}_{(j,\ell)})}$$

$$= -\sum_{i,j,k'} N_{i,j,k'}(\boldsymbol{A}_{(i,k')} + \boldsymbol{B}_{(j,k')}) + \sum_{i,j} N_{i,j} \log \left(\sum_\ell \exp(\boldsymbol{A}_{(i,\ell)} + \boldsymbol{B}_{(j,\ell)})\right)$$

$$= -\sum_{i,j,k'} N_{i,j,k'} \left((\boldsymbol{W}^M \boldsymbol{W}^T)_{(i,k')} + (\boldsymbol{W}^V \boldsymbol{W}^T)_{(j,k')}\right)$$

$$+ \sum_{i,j} N_{i,j} \log \left(\sum_\ell \exp\left((\boldsymbol{W}^M \boldsymbol{W}^T)_{(i,\ell)} + (\boldsymbol{W}^V \boldsymbol{W}^T)_{(j,\ell)}\right)\right).$$

We define $\hat{P}_{i,j}(k')$ as the softmax probability—under current model parameters—that, given current node $i$ and target node $j$, the model predicts node $k'$ as the token at step $n+2$:

$$\widehat{P}_{i,j}(k') = \frac{\exp\left(\boldsymbol{A}_{(i,k')} + \boldsymbol{B}_{(j,k')}\right)}{\sum_\ell \exp\left(\boldsymbol{A}_{(i,\ell)} + \boldsymbol{B}_{(j,\ell)}\right)}.$$

Then we have that the total gradient is the sum of contributions from all pairs $(i,j)$:

$$\frac{\partial \ell^{(2)}(\mathcal{D})}{\partial \boldsymbol{W}^T_{(d,k')}} = \sum_{i,j} \underbrace{\left[ \left( \widehat{P}_{i,j}(k') - P^{\text{data}}_{i,j}(k') \right) \cdot N_{i,j} \cdot \left( \boldsymbol{W}^M_{(i,d)} + \boldsymbol{W}^V_{(j,d)} \right) \right]}_{\text{Contribution from pair } (i,j)}. \qquad (15)$$

The proof of Theorem 1 follows by analyzing the contribution of each term in this summation. For any specific pair $(i,j)$ with $N_{i,j} > 0$, the contribution is determined by the prediction error.

Similarly, the gradient with respect to $\boldsymbol{W}^M$ is derived as the sum of contributions from all applicable pairs $(j,k')$:

$$\frac{\partial \ell^{(2)}(\mathcal{D})}{\partial \boldsymbol{W}^M_{(i,k)}} = \sum_{j,k'} \underbrace{\left[ \left( \widehat{P}_{i,j}(k') - P^{\text{data}}_{i,j}(k') \right) \cdot N_{i,j} \cdot \boldsymbol{W}^T_{(k,k')} \right]}_{\text{Contribution from pair } (j,k')}. \qquad (16)$$

The proof of Theorem 2 for the backbone parameter $\boldsymbol{W}^M$ follows from analyzing the sign of each term in this summation. For any term in this sum that corresponds to a context $(i,j)$ with $N_{i,j} > 0$, its sign is determined by the prediction error for that context.

The total gradient for $\boldsymbol{W}^M_{(i,k)}$ is the sum of all such positive and negative contributions over all applicable pairs $(j,k')$. As stated in the theorem, analogous reasoning holds for gradients w.r.t. $\boldsymbol{W}^V$:

$$\frac{\partial \ell^{(2)}(\mathcal{D})}{\partial \boldsymbol{W}^V_{(j,k)}} = \sum_{i,k'} \underbrace{\left[ \left( \widehat{P}_{i,j}(k') - P^{\text{data}}_{i,j}(k') \right) \cdot N_{i,j} \cdot \boldsymbol{W}^T_{(k,k')} \right]}_{\text{Contribution from pair } (i,k')}. \qquad (17)$$

This concludes the proof.

## C.2 Properties of Converged Solutions

Wang et al. (2024b) analyzed the autoregressive learning, and the gradient of the next-token prediction loss with respect to the parameter $\boldsymbol{W}^M$ is given. Let the model's predicted distribution for each context $(i,j)$ be

$$\hat{P}^{(1)}_{i,j}(k) = \frac{\exp\left( \boldsymbol{W}^M_{(i,k)} + \boldsymbol{W}^V_{(j,k)} \right)}{\sum_\ell \exp\left( \boldsymbol{W}^M_{(i,\ell)} + \boldsymbol{W}^V_{(j,\ell)} \right)}.$$

Let $N_{i,j,k}$ denote the number of times in $\mathcal{D}$ that (a) the current node is $i$, (b) the attention target is $j$, and (c) the next token is $k$. Define the empirical distribution $P^{\text{data}(1)}_{i,j}(k) = \frac{N_{i,j,k}}{N_{i,j}}$ $(N_{i,j} > 0)$.

Then the gradient can be written as

$$\frac{\partial \ell^{(1)}(\mathcal{D})}{\partial \boldsymbol{W}^M_{(i,k)}} = \sum_j \underbrace{\left[ \left( \hat{P}^{(1)}_{i,j}(k) - P^{\text{data}(1)}_{i,j}(k) \right) \cdot N_{i,j} \right]}_{\text{Contribution from } j}. \qquad (18)$$

Analogous results hold for gradients w.r.t. $\boldsymbol{W}^V$.

At convergence, the gradient vanishes, i.e., $\frac{\partial \ell^{(1)}(\mathcal{D})}{\partial \boldsymbol{W}^M_{(i,k)}} = 0$, which implies that for all contexts, the model predicted distribution matches the empirical distribution, $\hat{P}^{(1)}_{i,j}(k) = P^{\text{data}(1)}_{i,j}(k)$. This indicates that upon convergence, for each context $(i,j)$, the predicted probability distribution of the next token coincides with the frequency observed in the dataset.

Similarly, when the two-step loss $\ell^{(2)}$ has a zero gradient with respect to the parameter $\boldsymbol{W}^M$, the model's predicted distribution for the next two steps still matches the empirical distribution: $\hat{P}^{(2)}_{i,j}(k) = P^{\text{data}(2)}_{i,j}(k)$, which ensures that multi-step predictions remain consistent with the empirical distribution.

We further analyze the properties of the solution for this simplified model. Taking the logarithm of the one-step predicted distribution at convergence gives

$$\boldsymbol{W}^M_{(i,k)} + \boldsymbol{W}^V_{(j,k)} = \log P^{\text{data}(1)}_{i,j}(k) + c_{i,j}, \quad c_{i,j} = \log \sum_\ell \exp(\boldsymbol{W}^M_{(i,\ell)} + \boldsymbol{W}^V_{(j,\ell)}) \qquad (19)$$

where $c_{i,j}$ is the normalization constant.

It can be seen that for each context $(i, j)$, the sum of the weights $\boldsymbol{W}^M_{(i,k)} + \boldsymbol{W}^V_{(j,k)}$ is uniquely determined, but the individual matrices $\boldsymbol{W}^M$ and $\boldsymbol{W}^V$ are not unique, since one can add a constant to $\boldsymbol{W}^M$ and subtract the same constant from $\boldsymbol{W}^V$ without changing the Softmax output.

By the same reasoning, for the 2nd-step prediction loss $\ell^{(2)}$, the sum of weights satisfies

$$(\boldsymbol{W}^M \boldsymbol{W}^T)_{(i,k)} + (\boldsymbol{W}^V \boldsymbol{W}^T)_{(j,k)} = \log P^{\text{data}(2)}_{i,j}(k) + c_{i,j},$$

$$c_{i,j} = \log \sum_\ell \exp\left( (\boldsymbol{W}^M \boldsymbol{W}^T)_{(i,\ell)} + (\boldsymbol{W}^V \boldsymbol{W}^T)_{(j,\ell)} \right), \quad (20)$$

and due to the highly non-convex nature of the loss with respect to these matrices and the mutual coupling of gradient terms across different node pairs, the model may admit multiple equivalent parameter factorizations. Consequently, the individual parameters $\boldsymbol{W}^M$, $\boldsymbol{W}^V$, and $\boldsymbol{W}^T$ are not unique, while the multi-step predictions determined by their combined effect remain consistent with the empirical distribution.

From the above analysis, we conclude that the Softmax output probability distribution is uniquely determined at convergence, while the individual weight matrices are not. The model captures the transition statistics of the data as it gradually converges.

## D  EXPERIMENTS ON DIRECTED GRAPHS WITH CYCLES

**Experimental Setup:** In each experiment, we generate a random directed graph with $n = 100$ nodes, where every ordered pair $(i, j)$ with $i \neq j$ is included as a directed edge independently with probability $p = 0.05$. We compute the transitive closure of the graph to identify all reachable source–target pairs $(s, t)$ and, for each reachable pair, sample up to $m = 20$ paths using a randomized depth-first search with a length cap (no node revisits are allowed). To restrict the observable reachability matrix and to construct degree-2 test paths, we set this maximum path length to 6. We assign 10% of the $(s, t)$ pairs to the training set and the remaining 90% to the test set; all one-hop edges $(s, t) \in \mathcal{E}$ are always included in the training set as direct paths. The model architecture and all other training configurations follow Section 5.1.

**Results:** As shown in Table 6, MTP models achieve substantial accuracy gains over the 1-Token Prediction (NTP) baseline on degree-2 tests. This improvement persists with or without NTI and Transformer-based Transfer Layers, demonstrating that multi-token supervision significantly enhances the model's ability to learn transitive reachability.

## E  FURTHER WEIGHT ANALYSIS OF SIMPLIFIED MODEL

Fixing the transfer layer $\boldsymbol{W}^T$ to the ground-truth adjacency, we train the simplified model on a 20-node graph.

Table 6: **Path accuracy (%) on degree-2 paths in 100-node cyclic graphs.** Metrics include graph-level accuracy (with $\pm$ standard error) and path-level accuracy. Results are averaged over 100 graphs.

| MODEL | Graph $\pm$ / Path |
| --- | --- |
| 1-Token (baseline) (Wang et al., 2024b) | $59.89_{\pm 0.52}$ / 59.49 |
| 2-Token (Meta's) (Gloeckle et al., 2024) | $67.70_{\pm 0.45}$ / 67.88 |
| 2-Token + NTI (linear transfer layer) | $69.37_{\pm 0.46}$ / 69.10 |
| 2-Token + 1-layer Transformer | $68.12_{\pm 0.42}$ / 68.12 |
| 2-Token + NTI + 1-layer | $\mathbf{69.79_{\pm 0.40}}$ / **69.65** |
| 3-Token (Meta's) (Gloeckle et al., 2024) | $64.13_{\pm 0.42}$ / 63.90 |
| 3-Token + NTI (linear transfer layer) | $65.27_{\pm 0.44}$ / 65.10 |
| 3-Token + 1-layer Transformer | $64.86_{\pm 0.43}$ / 65.00 |
| 3-Token + NTI + 1-layer | $\mathbf{66.68_{\pm 0.42}}$ / **66.67** |

Figure 9 shows the learned weight matrices after training with 1-Token Prediction. As can be seen, the model learns the adjacency relations in $\boldsymbol{W}^M$ and the observed reachability relations in $\boldsymbol{W}^V$. Figure 10 shows the learning results of 2-Token Prediction, where the matrices contain entries that are theoretically learned under the $\ell^{(2)}(\mathcal{D})$ constraint. These entries have weights slightly higher than the background, but remain significantly weaker than those reinforced by $\ell^{(1)}(\mathcal{D})$. Ultimately, spurious adjacency relations are observed in $\boldsymbol{W}^M$, while $\boldsymbol{W}^V$ partially captures unobserved reachability relations.

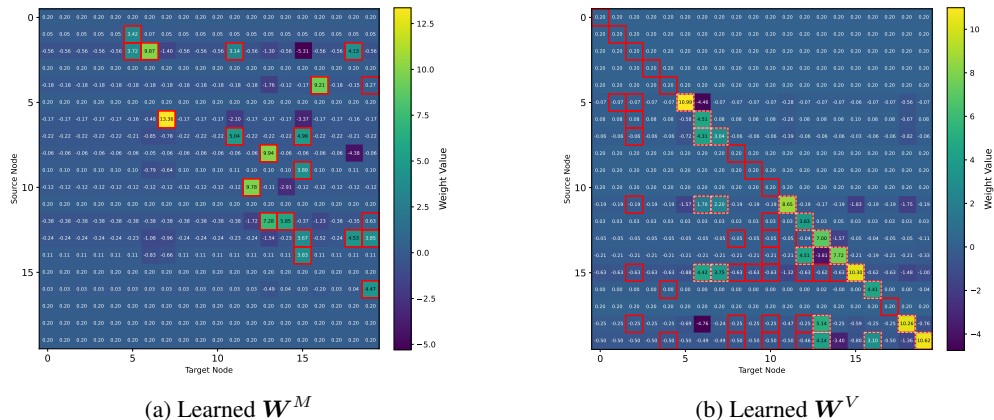

(a) Learned $\boldsymbol{W}^M$  (b) Learned $\boldsymbol{W}^V$

Figure 9: **Weight visualizations on a 20-node graph trained with 1-Token Prediction.** Red boxes highlight true adjacency in $\boldsymbol{W}^M$ (left) and true reachability in $\boldsymbol{W}^V$ (right). In the right plot, light dashed boxes indicate observed reachability.

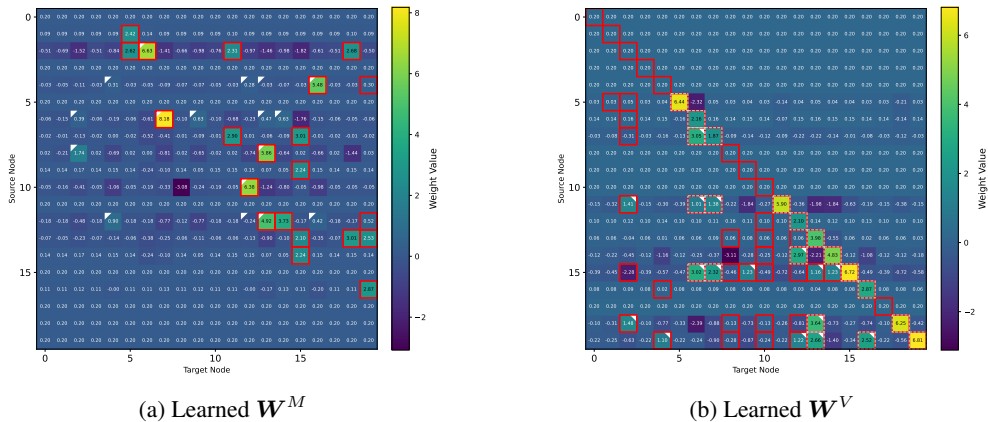

(a) Learned $\boldsymbol{W}^M$  (b) Learned $\boldsymbol{W}^V$

Figure 10: **Weight visualizations on a 20-node graph with fixed transfer layer $\boldsymbol{W}^T$ trained with 2-Token Prediction.** Red boxes highlight true adjacency in $\boldsymbol{W}^M$ (left) and true reachability in $\boldsymbol{W}^V$ (right). In the right plot, light dashed boxes indicate observed reachability. White triangles indicate theoretically learnable entries under $\ell^{(2)}(\mathcal{D})$.

To rigorously validate our theoretical analysis, Figure 6 in the main text shows the training results of 2-Token Prediction on a 100-node graph when the transfer layer $\boldsymbol{W}^T$ is fixed to the ground-truth adjacency matrix. However, in the general 2-Token Prediction setting, the transfer layer $\boldsymbol{W}^T$ is learned, with its input being the backbone-predicted next-step logits rather than the true next-step one-hot vectors. As a result, it exhibits certain deviations from the ground-truth adjacency matrix.

Figure 11 shows the training results of general 2-Token Prediction. The behavior of $\boldsymbol{W}^M$ is similar to the previous analysis: the true adjacency relations learned via $\ell^{(1)}(\mathcal{D})$ remain prominent. Since $\boldsymbol{W}^T$ is not a perfect ground-truth adjacency matrix, $\boldsymbol{W}^V$ not only captures the theoretically learnable reachability relations under $\ell^{(2)}(\mathcal{D})$, but also learns additional high-weight reachability

relations, which also belong to the true reachability in the graph. Table 7 shows the average weight statistics.

To explain this phenomenon in a more general way, training paths often contain sequences that differ only by additional intermediate nodes. As a result, the model tends to assign relatively high logits to tokens corresponding to these intermediate nodes, which can cause the transfer layer to learn transitions that do not exist in the true adjacency. This represents common noise in the transfer layer. At the same time, due to the structure of the paths, the transfer layer can propagate reachability through multiple steps, enabling it to capture additional correct reachability relations. Consequently, the resulting reachability relations learned by $\boldsymbol{W}^V$ include both theoretically learnable entries and other valid relations present in the graph.

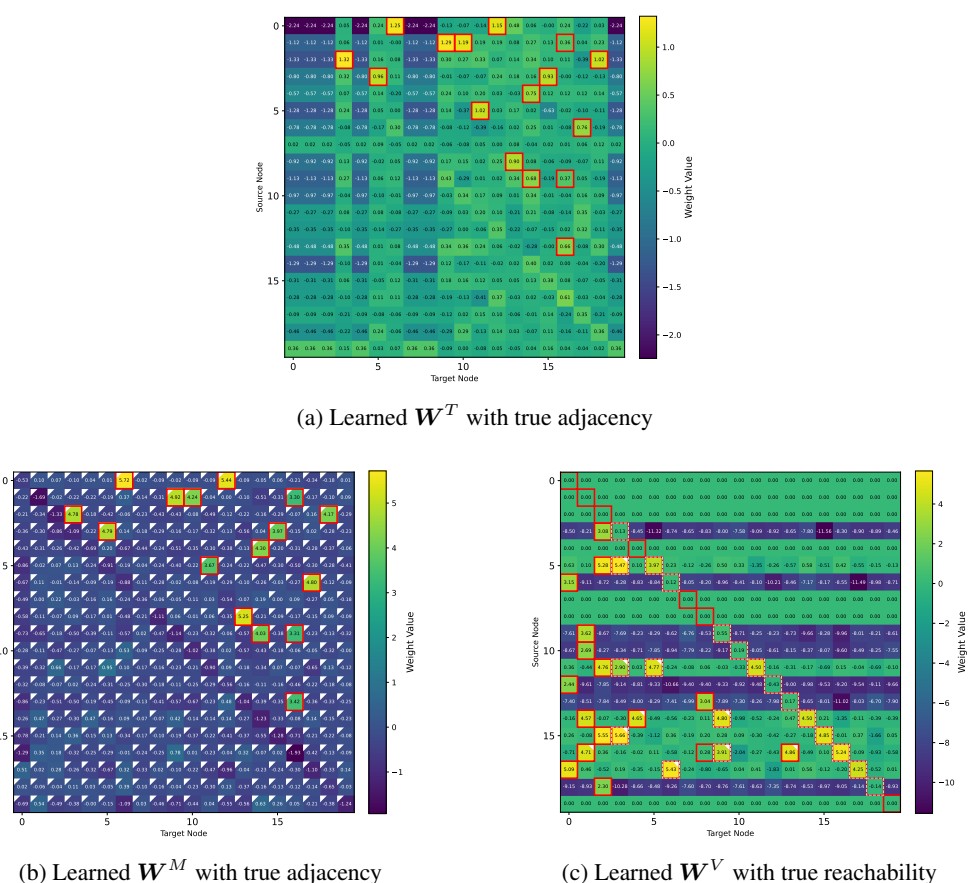

(a) Learned $\boldsymbol{W}^T$ with true adjacency

(b) Learned $\boldsymbol{W}^M$ with true adjacency

(c) Learned $\boldsymbol{W}^V$ with true reachability

Figure 11: **Weight visualizations on a 100-node graph under general 2-Token Prediction for the first 20 nodes.** (a) $\boldsymbol{W}^T$ with true adjacency highlighted in red boxes. (b) $\boldsymbol{W}^M$ with true adjacency in red boxes and theoretically learnable adjacency (under $\ell^{(2)}(\mathcal{D})$) marked by white triangles: extra entries are incorrect. (c) $\boldsymbol{W}^V$ with true reachability in red boxes, observed reachability in light dashed boxes, and theoretically learnable reachability in white: extra entries are correct and unobserved.

This process allows the model to capture a broader set of correct reachability relations. As a result, the 2-Token Prediction models continue to achieve strong performance on the degree-3 paths in the test set, as reported in Table 1.

# F  BLOCKSWORLD EXPERIMENTAL SETUP

The Blocksworld benchmark (Valmeekam et al., 2023) is a well-known task in classical planning. It involves a set of colored blocks, where each block can be either placed on the table or stacked on

Table 7: Average weights of different entry types in $\boldsymbol{W}^M$ and $\boldsymbol{W}^V$ in the 100-node graph under general 2-Token Prediction in the simplified model.

| MATRIX | TYPE | VALUE |
|---|---|---|
| $\boldsymbol{W}^M$ | True adjacency | 2.57 |
| | Theoretically learnable | 0.13 |
| | Other entries | -0.11 |
| $\boldsymbol{W}^V$ | Observed reachability | 0.89 |
| | True but not observed reachability | 0.31 |
| | Theoretically learnable | 0.52 |
| | True but not observed & theoretically learnable reachability | 0.61 |
| | Other entries (not true reachability) | -1.21 |

top of another block. The objective is to plan a sequence of valid actions that transform an initial configuration into a goal configuration.

We consider a version with 4 blocks. This setup results in a complete state transition graph containing 73 nodes, where each node represents a unique, valid block configuration, and each edge $(u, v)$ denotes a legal atomic action that transitions the system from state $u$ to state $v$. This task can therefore be seen as an instance of the path-planning problem described in the main text, but on a fixed, predefined graph. For example, node 49 represents the state where block A is on the table, B is on C, C is on D, and D is on the table.

To simulate a more general planning scenario where the model does not see the full state space during training, we use only a small subset of state transition processes to construct the training set. Each training example corresponds to a valid path sampled from the graph. The test set is composed of randomly selected source–target state pairs, and the model is required to generate the sequence of intermediate states connecting them, i.e., a valid path.

**Dataset Construction:** The training set includes all one-hop edges $(s, t) \in \mathcal{E}$ added as direct paths to ensure learning of adjacency relations. For each path length from 2 to 6, we sample $n$ acyclic paths, where $n$ is varied as 100, 200, 300, 400, or 500 to create training sets of different sizes. Each path is formatted as "s t s a ... t \n".

The test set is fixed and consists of 5,000 randomly sampled paths with lengths greater than 1. For each training size, all models are trained on the same set of paths and evaluated on this same test set.

## G USE OF LARGE LANGUAGE MODELS

We acknowledge the use of Large Language Models (LLMs) in the preparation of this manuscript. Their role was strictly limited to assisting with and refining the writing. All research work in this paper, including problem formulation, theoretical design, experiment execution, data analysis, and conclusion derivation, was conducted independently by the human authors. After completing the initial draft, we used an LLM to check grammar, optimize sentence structure, and suggest academic terminology. All suggestions were reviewed and approved by the authors to ensure that the academic intent and scientific accuracy were preserved.

Therefore, the intellectual ownership, academic content, and final wording of this manuscript rest entirely with the human authors, who bear full responsibility for the originality and correctness of the work.

