# OpenReview forum: "Understanding and Enhancing the Planning Capability of Language Models via Multi-Token Prediction"
_ICLR.cc/2026/Conference — Submitted to ICLR 2026_

### Official Review · Reviewer_qfv3 · 2025-10-18

**Soundness:** 3
**Presentation:** 2
**Contribution:** 2
**Rating:** 2
**Confidence:** 4

**Summary:**

This paper proposes to use multi-token predictions to improve language models' "planning" capabilities (e.g. path finding), which is an example of learning transitive relations.

Specifically, in addition to standard next-token prediction, they use a _transfer layer_ to predict a few future tokens; the same transfer layer is shared across different future positions.
- Most of the results are on a graph path finding setup following ALPINE (Wang et al. 24), where the model is expected to learn the true reachability matrix (i.e. whether a pair of nodes are connected) given paths observed in the training set.

For theoretical results, they show with a simplified model that how predicting two-steps head (rather than only the next step) affects the model weights, in terms of the signs of gradients.

For empirical results, they apply multi-step prediction to improve the accuracy on the above synthetic path finding task and Blocksworld.
As implementation details, they use:
- next-token injection, which feeds in the ground truth tokens as input during training.
- a Transformer layer as the transfer layer, where the self-attention is across different dimensions.

**Strengths:**

- This paper applies multi-token prediction to improve the transitive reasoning abilities in language models, which is well-justified.
- The paper discusses both the pros and cons of multi-token prediction: while multi-token prediction helps capture higher-order reachability (Theorem 1,2), the paper also mentions that it may incorrectly bias the 1-step transition probability.
  - The paper claims the benefit of improving transitive reachability outweighs the risk of biasing 1-step transitions.
- The empirical results are promising:
  - For path finding, 2-token or 3-token prediction improves the accuracy especially on paths that require composing information from more than 2 training data sequences.
  - For Blockworld, the paper shows that multi-token prediction provides some improvement over the 1-token baseline at different path lengths.
- The paper provides mechanistic study on how multi-token prediction affects the learned model.

**Weaknesses:**

I'm concerned about insufficient contribution.

- Theorem 1 and 2 (the results are about the signs of per-step gradients on a much simplified model) do not offer much beyond the intuitive explanation that the weight matrices are made to capture ground truth transitions.
  - This also comes with the cost of introducing notations (that are unnecessary for getting the key messages of the section) that impedes reading, so I'd suggest to keep the informal theorem in the main paper and move the details to the appendix.

- The experiment results are too limited to be convincing.
  - Multi-token prediction aligns well with the structure of the path finding task, so improved performance is expected. For Blocksworld, the amount of improvement is minimal, and even worse than the baseline when using a linear transfer layer. The paper does not demonstrate how much multi-token prediction would benefit more general setup.
  - For the path finding setup, the paper only considers Erdos graphs (with p=0.1) and not more structured graphs which are closer to applications (e.g. stochastic block models).

**Questions:**

- Sec 5.2, weight analysis: My understanding is that the transfer layer $W^T$ is not a Transformer layer in this case, since otherwise projecting $W^T$ doesn't make sense. Please clarify this in the writing.
- Table 3: It's not clear how to compare the value of "0.82" and "4.01". What are the ranges of the values? What's the variance in these values when averaged over several runs? Are normalization layer used? If yes, are these values before or after normalization layers?
- Table 1, 5: how many layers are used for the 1-token baseline? If it's 1-layer 1-head, then it's unclear whether the gain from multi-token prediction is due to the use of more Transformer layers (as transfer layers), or from multi-token training.

Minor: on line 112: has $N$ been defined?

---

> ### Author Response · Authors · 2025-11-18
> **Official Comment by Authors (1/4)**
>
> > **W**: I'm concerned about insufficient contribution.
>
> **A**: We appreciate the reviewer’s concerns regarding the paper’s contributions. Rather than aiming for task-specific performance gains, our work focuses on a theoretical examination of how Multi-Token Prediction (MTP) enables planning at the token level, thereby explaining the core mechanism by which Transformers learn transitive relations under MTP. Transitive planning has long been a bottleneck for Transformer-based models [1], and existing studies in MTP offer little clarity on the internal learning process behind it, and most remains at the experimental level [2-5].
>
> We provide a novel analysis of the learning dynamics of MTP, demonstrating how MTP overcomes the inherent limitations of single-token prediction in capturing transitive relations, and highlighting its potential to offer theoretical insights for MTP-related research. At the same time, we empirically validate our analysis.
>
> We believe that this mechanistic understanding offers both theoretical and empirical support for extending MTP to more complex planning and reasoning scenarios in future work. We hope this study encourages the community to combine theoretical and empirical approaches when studying LLMs, advancing their planning capabilities and structural comprehension.
>
> [1] Zhang et al. *Transformer-Based Models Are Not Yet Perfect At Learning to Emulate Structural Recursion*, TMLR 2024.
> [2] Thankaraj et al. *Looking beyond the next token*, arXiv:2504.11336.
> [3] Liu et al. *Leap multi-token prediction beyond adjacent context for large language models*, NeurIPS 2025.
> [4] Mahajan et al. *Beyond Multi-Token Prediction: Pretraining LLMs with Future Summaries*, arXiv:2510.14751.
> [5] Pal et al. *Future Lens: Anticipating Subsequent Tokens from a Single Hidden State*, ACL 2023
>
>
> > **W1**: Theorem 1 and 2 (the results are about the signs of per-step gradients on a much simplified model) do not offer much beyond the intuitive explanation that the weight matrices are made to capture ground truth transitions.
> This also comes with the cost of introducing notations (that are unnecessary for getting the key messages of the section) that impedes reading, so I'd suggest to keep the informal theorem in the main paper and move the details to the appendix.
>
> **A1**: Regarding the comments on Theorems 1 and 2, these theorems serve to provide an intuitive explanation of Multi-Token Prediction (MTP) while also revealing patterns of gradient dynamics. At the same time, they also reveal the patterns of gradient directions and weight dynamics under multi-token supervision, which are not immediately apparent from intuition alone. These patterns provide useful guidance for model design and optimization and support our core claims. Following the suggestions, we have simplified the theorems: we reduced the mathematical notation and added more descriptive explanations, and moved the detailed derivations and formal expressions to the appendix.
>
> Revision part (line: 213–219, 233–239):
>
> **Theorem 1.** For any pair $(i, j)$ in dataset $\mathcal{D}$ with $N\_{i,j} > 0$, let $P^{\mathrm{data}}\_{i,j}(k') = N\_{i,j,k'}/N\_{i,j}$ denote the empirical probability of the second-next node $k'$. The contribution of each $(i, k')$ pair to the gradient $\frac{\partial \ell^{(2)}(\mathcal{D})}{\partial W^{T}\_{(d,k')}}$ is proportional to the prediction error $\widehat{P}\_{i,j}(k') - P^{\mathrm{data}}\_{i,j}(k')$, the sample count $N\_{i,j}$, and the combined weight $(W^{M}\_{(i,d)} + W^{V}\_{(j,d)})$. When the contribution is positive, it pushes the weight $W^{T}\_{(d,k')}$ to **decrease**; when negative, it pushes the weight to **increase**. The total gradient is the sum of contributions over all pairs $(i, j)$ in the dataset.
>
> **Theorem 2.** For any pair $(i, j)$ in dataset $\mathcal{D}$ with $N\_{i,j} > 0$, let $P^{\mathrm{data}}\_{i,j}(k') = N\_{i,j,k'}/N\_{i,j}$ denote the empirical probability of the second-next node $k'$. The contribution of each $(\text{current node } i, \text{second-step node } k')$ pair to the gradient $\frac{\partial \ell^{(2)}(\mathcal{D})}{\partial W^{V}\_{(j,k)}}$ is proportional to the prediction error $\widehat{P}\_{i,j}(k') - P^{\mathrm{data}}\_{i,j}(k')$, the sample count $N\_{i,j}$, and the weight $W^{T}\_{(k,k')}$. When the contribution is positive, it pushes the weight $W^{V}\_{(j,k)}$ to **decrease**; when negative, it pushes the weight to **increase**. The total gradient is the sum of contributions over all pairs $(i, k')$ in the dataset.

---

> ### Author Response · Authors · 2025-11-18
> **Official Comment by Authors (2/4)**
>
> > **W2**: The experiment results are too limited to be convincing.
>
> > **W2.1**: Multi-token prediction aligns well with the structure of the path finding task, so improved performance is expected. For Blocksworld, the amount of improvement is minimal, and even worse than the baseline when using a linear transfer layer. The paper does not demonstrate how much multi-token prediction would benefit more general setup.
>
> **A2.1**:  Path planning is a fundamental abstraction for many complex planning tasks, and standard Transformers face a bottleneck in the “transitive planning” required for these tasks. Our goal is to use Multi-Token Prediction (MTP) to reveal the mechanism that overcomes this bottleneck, rather than to choose a task where MTP’s success is naturally expected. The experiments validate our theoretical analysis: enhanced MTP achieves higher accuracy than standard MTP and establishes a higher performance ceiling compared to the baseline.
>
> For Blocksworld, noise in the backbone limits full-model optimization (lines 300–303: Its performance is constrained by the backbone output ${h}_n$’s ability to predict the next node, where the deviation between the predicted and true next step introduces noise.). As learning is enhanced, this noise is also amplified; however, when the task itself does not require such additional reasoning capability, the disadvantage introduced by this noise becomes even more pronounced, so the original MTP does not necessarily outperform the baseline. This motivates the design of our improved MTP variants. Overall, MTP provides an explicit pathway for learning transitive reachability.
>
> The paper focuses on the theoretical foundations of how Transformers process interactions between tokens and on analyzing their planning capability, providing a solid and generalizable framework. Our goal is to make the transitive reasoning mechanism explicit and theoretically grounded, which we believe is a necessary step before scaling MTP to broader domains. Future work will further investigate extending the approach to applications (lines 525–528: Future work may explore applying Multi-Token Prediction to domains such as visual navigation, robotic manipulation, or text-based planning by incorporating appropriate state discretization or structural extraction methods.)
>
>
>
> > **W2.2**: For the path finding setup, the paper only considers Erdos graphs (with p=0.1) and not more structured graphs which are closer to applications (e.g. stochastic block models).
>
> **A2.2**:  We conducted an additional experiment on a single structured graph generated by a stochastic block model (SBM). The graph contains 100 nodes divided into 4 equally-sized communities, with an intra-community edge probability of 0.1 and an inter-community edge probability of 0.01. The task remains degree-2 path prediction, consistent with our main experiments.
> The results (path-level accuracy) are as follows:
>
> | Model | Path Accuracy (%) |
> |--|--|
> | 1-Token (baseline) | 60.90 |
> | 2-Token (Meta’s) | 68.30 |
> | 2-Token + NTI | 69.85 |
> | 2-Token + 1-layer Transformer | 68.70 |
> | 2-Token + NTI + 1-layer | **70.55** |
> | 3-Token (Meta’s) | 65.50 |
> | 3-Token + NTI | 66.65 |
> | 3-Token + 1-layer Transformer | 66.10 |
> | 3-Token + NTI + 1-layer | **67.20** |
>
> It can be seen that these results converge with the results on Erdos graphs in Section 5.1.
>
> For enhancing the experiments part, we also add one experiment on directed cyclic graphs. The results are as follows (graph-level ± standard error / path-level accuracy for degree-2 paths, 100-node graphs, edge probability p=0.05):
>
> | Model | Accuracy |
> |---|---|
> | 1-Token (baseline) | 59.89 ± 0.52 / 59.49 |
> | 2-Token (Meta's) | 67.70 ± 0.45 / 67.88 |
> | 2-Token + NTI | 69.37 ± 0.46 / 69.10 |
> | 2-Token + 1-layer Transformer | 68.12 ± 0.42 / 68.12 |
> | 2-Token + NTI + 1-layer | **69.79 ± 0.40 / 69.65** |
> | 3-Token (Meta's) | 64.13 ± 0.42 / 63.90 |
> | 3-Token + NTI | 65.27 ± 0.44 / 65.10 |
> | 3-Token + 1-layer Transformer | 64.86 ± 0.43 / 65.00 |
> | 3-Token + NTI + 1-layer | **66.68 ± 0.42 / 66.67** |
>
> The experiment also validates the conclusions of the paper, and we have added it to Appendix D.
>
> We also want to point out that our theoretical findings are not relying on any specific structural property of the underlying graph, and thus it works for all types of graphs. Following the reviewer’s valuable suggestions, we have conducted experiments on other type of graphs and the results confirms the generality of our theoretical finding.

---

> ### Author Response · Authors · 2025-11-18
> **Official Comment by Authors (3/4)**
>
> > **Q1**：Sec 5.2, weight analysis: My understanding is that the transfer layer W^T is not a Transformer layer in this case, since otherwise projecting W^T doesn't make sense. Please clarify this in the writing.
>
> **A1**: We appreciate your keen observation and thank you for pointing out this ambiguity. Your understanding is correct.
>
> In Sec. 5.2, we mainly discuss the case where the transfer layer is linear. However, when the transfer layer is Transformer-based, it can also capture adjacency relationships and model interactions along the sequence dimension, making it suitable for taking the output sequence as input. We clarify this part in the revised manuscript to make it easier to understand.
>
> Revised version (line 423–426): In a 100-node path-planning task, we project the linear transfer layer $W^{T}$ onto the node representations via $W_{t} W^{T} W_{o}$ to evaluate its ability to learn transition relations...
>
> > **Q2**: Table 3: It's not clear how to compare the value of "0.82" and "4.01". What are the ranges of the values? What's the variance in these values when averaged over several runs? Are normalization layer used? If yes, are these values before or after normalization layers?
>
> **A2**:
> > It is not immediately clear how to compare the values "0.82" and "4.01".
>
> According to Equation 6:
> $\frac{\partial \ell^{(2)}(\mathcal{D})}{\partial W^T} = \left( W\_o \left( h\_n + k \, (W\_t)\_{(:, u\_{n+1})} \right) \right)^\top \left( \hat{p}\_{n+2} - e\_{n+2} \right)$,
> after NTI explicitly supervises the learning of $W^T$, we can observe a significant improvement from 0.82 to 4.01, which validates the theory.
>
> Furthermore, according to Theorem 2 (lines 233–239):
> For any pair $(i, j)$ in dataset $\mathcal{D}$ with $N\_{i,j} > 0$, let $P^{\mathrm{data}}\_{i,j}(k') = N\_{i,j,k'}/N\_{i,j}$ denote the empirical probability of the second-next node $k'$. The contribution of each $(\text{current node } i, \text{second-step node } k')$ pair to the gradient $\frac{\partial \ell^{(2)}(\mathcal{D})}{\partial W^V\_{(j,k)}}$ is proportional to the prediction error $\widehat{P}\_{i,j}(k') - P^{\mathrm{data}}\_{i,j}(k')$, the sample count $N\_{i,j}$, and the weight $W^T\_{(k,k')}$. When the contribution is positive, it pushes $W^V\_{(j,k)}$ to decrease; when negative, it pushes it to increase. The total gradient is the sum of contributions over all pairs $(i, k')$ in the dataset.
>
> A higher $W^T$ weight leads to larger updates in the backbone $W^M$ and $W^V$ layers, thereby reinforcing favorable gradient changes for the backbone model. This explains why the value 4.01 after NTI supervision reflects a stronger and more effective gradient signal compared to 0.82.
>
> > What are the ranges of the values?
>
> For the weights in Table 3, the ranges are as follows:
> - Without NTI: the values range from -0.85 to 1.38.
> - With NTI: the values range from -3.43 to 9.41.
>
> > What's the variance in these values when averaged over several runs?
>
> We conducted experiments over 10 different graphs and computed the average values of the mapped matrices. The resulting variances are as follows:
>
> | Condition | All entries | ADJ.AVG | NON-ADJ.AVG |
> |-----------|------------|---------|-------------|
> | No NTI | 0.008| 0.012| 0.004   |
> | NTI| 0.032  | 0.045| 0.020  |
>
> It can be seen that, although the specific weights vary across different graphs, the differences in their averages are small.
>
> > Are normalization layers used? If yes, are these values before or after normalization layers?
>
> No normalization were applied. There are several reasons for this choice:
> 1. Not normalizing allows us to more clearly observe the direct impact on the gradients, and thus the effect on the backbone model's updates (as in the first answer).
> 2. A few extreme values can strongly influence normalized numbers, which may obscure the actual differences in gradient magnitudes.
> 3. Regardless of NTI usage, the transfer layer naturally captures the one-step to two-step transition relations, and its values are driven by the training data. Therefore, normalization would not substantially affect the separability of the learned adjacency signals.
>
> For reference, we also provide the normalized values as follows:
>
> | Condition | ADJ.AVG | NON-ADJ.AVG | GAP |
> |-----------|-------|-----|------------|
> | No NTI    | 0.66 | 0.42  | 0.24  |
> | NTI       | 0.56| 0.26 | 0.30 |
>
> We will provide a clearer explanation in the revised manuscript.

---

> ### Author Response · Authors · 2025-11-18
> **Official Comment by Authors (4/4)**
>
> > **Q3**：Table 1, 5: how many layers are used for the 1-token baseline? If it's 1-layer 1-head, then it's unclear whether the gain from multi-token prediction is due to the use of more Transformer layers (as transfer layers), or from multi-token training.
>
> **A3**: To ensure a fair comparison, the 1-token baseline uses the same 1-layer, 1-head Transformer backbone as the multi-token models (line 362-363: All models use 120-dimensional embeddings and adopt a 1-layer, 1-head Transformer as the backbone.). The performance gain mainly comes from multi-token prediction itself, rather than an increase in parameters. As shown (lines 408-411: As shown in Figure 4, increasing backbone model complexity yields only marginal improvements. In contrast, the configuration combining 2-Token, NTI, and a Transformer-based Transfer Layer consistently yields stable accuracy gains.) and supported by [2], simply increasing the depth or width of the backbone provides limited improvement and does not overcome the inherent limitations of single-token prediction in learning transitive relations.
> We clarify this point in the revised manuscript (line 375-376: As shown in Figure 4, increasing backbone model complexity yields only slight changes (also shown in [6])).
>
> [6] Wang et al. *ALPINE: Unveiling The Planning Capability of Autoregressive Learning in Language Models*, NeurIPS 2024
>
> > Minor：on line 112: has N been defined?
>
> **A**: Yes, we miss the definition of $N$.
> Here, $N$ denotes the sequence length, and we clarify this in the paper. We have carefully proof-read the whole manuscript to avoid undefined notations.
>
> Revision (line 110–111):
> Given a training set $\mathcal{D}$ where $u = (u\_1, \dots, u\_N) \in \mathcal{D}$ is a valid path in the graph, **with $N$ denoting the sequence length**, we define the observed adjacency and reachability matrices in the observation graph $\mathcal{G}\_{\mathrm{obs}}$.

---

### Official Review · Reviewer_ynaJ · 2025-10-26

**Soundness:** 3
**Presentation:** 2
**Contribution:** 3
**Rating:** 6
**Confidence:** 4

**Summary:**

The paper studies how multi-token prediction (MTP) can improve learning of transitive relations for path planning with Transformers. It analyzes a simplified one-layer Transformer with a shared output head and a linear “transfer layer” $W^T$ used to predict tokens multiple steps ahead. The analysis argues that: (i) WT is updated by the multi-step loss to approximate adjacency; (ii) gradients backpropagated through $W^T$  encourage the backbone to encode higher-order reachability; and (iii) some spurious adjacency may also be introduced. Two training/architectural tweaks are proposed: Next-Token Injection (NTI) that injects the ground-truth next-token embedding into the transfer input during training, and a Transformer-based transfer layer. Experiments on synthetic DAG path planning and Blocksworld show improvements over next-token training and re-implementations of “Meta-style” and “DeepSeek-style” MTP.

**Strengths:**

1. The research question studied is critical and timely, with a clean framing (transitivity bottleneck in planning).

2. The paper proposes a practical training tweak (NTI) that reliably improves results

3. The evaluations are comprehensive, including multiple MTP depths, backbone variants, and scaling across graph sizes.

**Weaknesses:**

1. Theorems are proven under a highly stylized setting: no positional embeddings, identity embeddings and output projection, fixed and manually set attention that always attends to the target (second position), linear FFN, single head/layer, and linear $W^T$ acting on logits.

2. The result that “$W^T$ learns adjacency” depends critically on $(W^M_{(i,d)}+W^V_{(j,d)})>0$. This is not guaranteed; if the sum is negative (or zero), the update direction flips (or vanishes). The paper partially acknowledges this by conditioning statements on positivity but then informally equates “positive-correlated intermediate nodes” with feasible intermediates.

3. The statement that $W^T_{(n−1)}$ “approximates the (n−1)-th power of the adjacency matrix” is not established. There is no proof of convergence or identification, and in the general case $W^T$ can absorb both adjacency and reachability information due to the non-uniqueness of the factorization with $W_t$ and $W_o$.

4. Transformer-based transfer layer description is unclear. The paper says it “leverages self-attention to model dependencies across dimensions” of $h_n$. A standard Transformer attends over the sequence dimension; with a single vector $h_n$ as input, self-attention degenerates. If you reshape features into a length-d sequence, please specify the exact reshaping, positional encoding, number of heads, masking, and why this is preferable to a simple MLP. As written, this component is conceptually confusing and under-specified.

5. Experimental details are missing. Decoding strategy, optimizer, learning rate schedule, batch size, epochs/steps, temperature, label smoothing, and regularization are not stated in the main text. Although a code link is provided, there is only a README file in it.

6. Only DAGs are used. Real planning graphs often contain cycles. An evaluation on cyclic graphs (with appropriate sequence formatting to avoid trivial loops) would strengthen claims.

**Questions:**

Suggestions:

1. Some figures use very small fonts. Adjusting them could make it easier to read.
2. Validate the gradient-sign predictions empirically by logging the sign of parameter updates vs the (p̂−pdata) terms during training.

---

> ### Author Response · Authors · 2025-11-18
> **Official Comment by Authors (1/3)**
>
> > **W1**: Theorems are proven under a highly stylized setting: no positional embeddings, identity embeddings and output projection, fixed and manually set attention that always attends to the target (second position), linear FFN, single head/layer, and linear W^T acting on logits.
>
> **A1**:  We appreciate your comment. The settings used in our theorems are followed by common simplifications from prior work [1-3] to enable a clear and controlled theoretical analysis. Specifically:
> - Positional embeddings mainly affect the attention mechanism in this problem, and their effect can be observed in experiments;
> - Fixed and manually set attention, always attending to the target (second position), is also reflected in the experiments;
> - Identity embeddings and output projection are primarily used to simplify notation and make the derivations clearer;
> - Linear feedforward networks (FFN) and the removal of non-linearities are adopted to facilitate theoretical analysis, as done in many prior works.
>
> These assumptions allow us to clearly derive the theoretical patterns of gradient dynamics and weight evolution in Multi-Token Prediction (MTP) during the learning of transitive relations. The conclusions can still be validated experimentally in general Transformer models, indicating that the theoretical analysis captures the core mechanisms and provides valuable guidance for model design and improvement.
>
> [1] Bietti, A., *Birth of a Transformer: A Memory Viewpoint*, NeurIPS 2023.
> [2] Boix‑Adsera, E., Musatov, V., & Acharya, R., *Transformers Learn through Gradual Rank Increase*, NeurIPS 2023.
> [3] Yao, X., et al., *Enhancing In‑Context Learning Performance with Just SVD*, NeurIPS 2024.
>
>
> > **W2**: The result that “W^T learns adjacency” depends critically on (W^M_{(i,d)}+W^V_{(j,d)}) > 0. This is not guaranteed; if the sum is negative (or zero), the update direction flips (or vanishes). The paper partially acknowledges this by conditioning statements on positivity but then informally equates “positive-correlated intermediate nodes” with feasible intermediates.
>
> **A2**: Thank you for pointing this out. We acknowledge that the original wording was imprecise. In practice, the next-token loss encourages the model to assign higher scores to feasible intermediate nodes, leading to stronger updates.
> We have revised the theorem and main text for clarity.
>
> Revision parts:
>
> Theorem (line: 213–219, 233–239):
>
> **Theorem 1.** For any pair $(i, j)$ in dataset $\mathcal{D}$ with $N\_{i,j} > 0$, let $P^{\mathrm{data}}\_{i,j}(k') = N\_{i,j,k'}/N\_{i,j}$ denote the empirical probability of the second-next node $k'$. The contribution of each $(i, k')$ pair to the gradient $\frac{\partial \ell^{(2)}(\mathcal{D})}{\partial W^T\_{(d,k')}}$ is proportional to the prediction error $\widehat{P}\_{i,j}(k') - P^{\mathrm{data}}\_{i,j}(k')$, the sample count $N\_{i,j}$, and the combined weight $(W^M\_{(i,d)} + W^V\_{(j,d)})$. When the contribution is positive, it pushes the weight $W^T\_{(d,k')}$ to **decrease**; when negative, it pushes the weight to **increase**. The total gradient is the sum of contributions over all pairs $(i, j)$ in the dataset.
>
> **Theorem 2.** For any pair $(i, j)$ in dataset $\mathcal{D}$ with $N\_{i,j} > 0$, let $P^{\mathrm{data}}\_{i,j}(k') = N\_{i,j,k'}/N\_{i,j}$ denote the empirical probability of the second-next node $k'$. The contribution of each $(\text{current node } i, \text{ second-step node } k')$ pair to the gradient $\frac{\partial \ell^{(2)}(\mathcal{D})}{\partial W^V\_{(j,k)}}$ is proportional to the prediction error $\widehat{P}\_{i,j}(k') - P^{\mathrm{data}}\_{i,j}(k')$, the sample count $N\_{i,j}$, and the weight $W^T\_{(k,k')}$. When the contribution is positive, it pushes the weight $W^V\_{(j,k)}$ to **decrease**; when negative, it pushes the weight to **increase**. The total gradient is the sum of contributions over all pairs $(i, k')$ in the dataset.
>
> **Main text** (line: 234-238):
> ...thereby strengthening the $b \rightsquigarrow j$ relation. Therefore, for any node $k$, when ${W}^T_{(k,k')}$ captures the true adjacency relationship between $k$ and $k'$...

---

> ### Author Response · Authors · 2025-11-18
> **Official Comment by Authors (2/3)**
>
> > **W3**: The statement that W^T_{(n-1)} “approximates the (n−1)-th power of the adjacency matrix” is not established. There is no proof of convergence or identification, and in the general case W^T can absorb both adjacency and reachability information due to the non-uniqueness of the factorization with
> W_t and W_o.
>
> **A3**: Thank you for your comment. The concern is reasonable. To clarify, we provide a detailed answer from three perspectives:
> 1. The statement “derived from Theorems 1 and 2” indicates that the conclusion is based on a simplified model and then naturally extended to the original Transformer.
> 2. Due to the non-uniqueness of matrix factorization, we acknowledge that information corresponding to the $(n-1)$-th power of the adjacency matrix may flow into combinations of $W\_t$ and $W\_o$. However, prior work shows that the input embedding layer and output layer serve different functions in the model [4-5], so the transition information in $W^T$ is only partially diluted.
> 3. While $W^T$ can also absorb adjacency and reachability information, after mapping through $W\_t$ and $W\_o$ back to node representations, most of the information still aligns with the adjacency matrix (including higher powers).
>
> Taken together, we believe this approximation is reasonable and necessary for explanatory purposes. We therefore use the term “approximates” cautiously in the main text, and here, it refers to the simplified model.
>
> [4] Press, O., & Wolf, L., *Using the Output Embedding to Improve Language Models*, ACL 2017.
> [5] Chung, H., et al., *Rethinking Embedding Coupling in Pre-trained Language Models*, ICLR 2021.
>
> > **W4**: Transformer-based transfer layer description is unclear. The paper says it “leverages self-attention to model dependencies across dimensions” of h_n. A standard Transformer attends over the sequence dimension; with a single vector h_n as input, self-attention degenerates. If you reshape features into a length-d sequence, please specify the exact reshaping, positional encoding, number of heads, masking, and why this is preferable to a simple MLP. As written, this component is conceptually confusing and under-specified.
>
> **A4**: The original statement that the "Transformer-based transfer layer models dependencies across the dimensions of $h_n$" can indeed be misleading. In fact, our input is the entire sequence $H = [h_1, \dots, h_N]$, and self-attention operates along the sequence dimension. Due to the lack of a rigorous mathematical description, the original text was unclear. We have supplemented the revised manuscript (line 319-337) with detailed reasons to clarify why this approach is superior to a linear mapping. The summarzied reason is that the linear transfer layer maps each sequence step independently, limiting the flow of contextual information. By replacing it with a Transformer-based transfer layer, the model leverages self-attention to capture dependencies across sequence positions. As a result, each step’s gradient aggregates contributions from subsequent steps, allowing the backbone to fully utilize sequence context during training, which improves both performance and training stability.
>
> > **W5**: Experimental details are missing. Decoding strategy, optimizer, learning rate schedule, batch size, epochs/steps, temperature, label smoothing, and regularization are not stated in the main text. Although a code link is provided, there is only a README file in it.
>
> **A5**:  We apologize for the omission of detailed experimental settings and the missing code in the code link. The repository is now updated with full training and evaluation scripts, ensuring reproducibility. All models were trained for 100K steps with these settings: AdamW optimizer, cosine LR decay (3e-4 initial), batch size 64/GPU, weight decay 0.1, dropout 0.1, and no label smoothing. Decoding uses top-k (k=50) sampling with temperature 1.0.

---

> ### Author Response · Authors · 2025-11-18
> **Official Comment by Authors (3/3)**
>
> > **W6**: Only DAGs are used. Real planning graphs often contain cycles. An evaluation on cyclic graphs (with appropriate sequence formatting to avoid trivial loops) would strengthen claims.
>
> **A6**:We have added experiments on directed cyclic graphs. To avoid loops, paths are constructed without revisiting the same node, and the sequence length is limited to at most 6 to ensure a sufficient number of degree-2 test paths. The results are as follows (graph-level ± standard error / path-level accuracy for degree-2 paths, 100-node graphs, edge probability p=0.05):
>
> | Model | Accuracy |
> |-------|----------|
> | 1-Token (baseline) | 59.89 ± 0.52 / 59.49 |
> | 2-Token (Meta's) | 67.70 ± 0.45 / 67.88 |
> | 2-Token + NTI | 69.37 ± 0.46 / 69.10 |
> | 2-Token + 1-layer Transformer | 68.12 ± 0.42 / 68.12 |
> | 2-Token + NTI + 1-layer | **69.79 ± 0.40 / 69.65** |
> | 3-Token (Meta's) | 64.13 ± 0.42 / 63.90 |
> | 3-Token + NTI | 65.27 ± 0.44 / 65.10 |
> | 3-Token + 1-layer Transformer | 64.86 ± 0.43 / 65.00 |
> | 3-Token + NTI + 1-layer | **66.68 ± 0.42 / 66.67** |
>
> The experiment also validates the conclusions of the paper, and we have added it to Appendix D.
>
>
> > **Q1**：Suggestions:
>
> > **Q1.1**: Some figures use very small fonts. Adjusting them could make it easier to read.
>
> **A1.1**: We apologize for the poor readability caused by the small fonts in some figures. We have adjusted the font sizes wherever possible without exceeding the space limits.
>
> > **Q1.2**：Validate the gradient-sign predictions empirically by logging the sign of parameter updates vs the (p̂−pdata) terms during training.
>
> **A1.2**: Thank you for the valuable suggestion. We agree that logging the sign of parameter updates against the $(\hat{p} - p_{\text{data}})$ terms during training could further strengthen the empirical support for our theoretical analysis.
>
> According to Theorem 2, the gradient of a specific element of $W^V$ (e.g., the $(j,k)$ entry) is determined by the aggregate contributions of all node pairs $(i, k')$ in the dataset. For each node pair $(i,k')$, the prediction error $\hat{P}\_{i,j}(k') - P^{\text{data}}\_{i,j}(k')$ is different, and the corresponding count $N_{i,j}$ also varies. Thus, the update direction of a single parameter is not determined by the sign of any single node pair, but by the combined contributions of all node pairs, which makes such a sign-alignment experiment difficult to carry out in practice.
>
> From the results perspective, in Sec. 5.2 we visualize the learned $W^M$ and $W^V$ matrices. The newly emerged relational patterns (i.e., newly highlighted entries) are fully consistent with the directions predicted by our theory, which directly validates the theoretical conclusions. We again thank the reviewer for the constructive feedback regarding the rigor of the theoretical analysis.

---

### Official Review · Reviewer_7Rg3 · 2025-11-01

**Soundness:** 4
**Presentation:** 4
**Contribution:** 4
**Rating:** 8
**Confidence:** 5

**Summary:**

The paper shows that making LLMs predict multiple future tokens instead of just the next one helps them plan better on graph-like tasks.
They prove that this setup makes the model’s extra prediction layer basically learn multi-hop adjacency so it actually understands longer paths.

**Strengths:**

- Novel loss function to better train the model. Under MTP the transfer layer aligns with multi-hop adjacency, thus directly tying weights to structure.

- Practical tweaks (Next-Token Injection and a Transformer-based transfer layer) are well-motivated and easy to slot into existing stacks.

- Measurable gains on harder generalization (degree-2/3 paths) and Blocksworld, not just easy cases.

- Learned transfer matrices progressively approximate true adjacency, thus giving a readable handle on structure.

**Weaknesses:**

- Scope is narrow: focus on DAG path-planning; little evidence for labeled/heterogeneous graphs or richer semantics (node/edge features).

- Scaling is lacking: experiments appear mid-scale; limited analysis for larger graphs, long paths, or real-world distributions.

- Lack of baselines: The experiments only compare against standard next-token training. There’s no evaluation versus other planning-aware or structure-inducing methods. So it’s unclear whether MTP is uniquely effective or just another form of multi-step supervision.

**Questions:**

- Permutation robustness: How stable are the learned transfer matrices under node relabeling or shuffled IDs? Any invariance tricks or results?

- Can the approach handle weighted/noisy edges or node/edge features (e.g., typed relations)? What changes in the theory?

---

> ### Author Response · Authors · 2025-11-18
> **Official Comment by Authors (1/3)**
>
> > **W1**: Focus on DAG path-planning; little evidence for labeled/heterogeneous graphs or richer semantics (node/edge features).
>
> **A1**:We appreciate your careful review and insightful comments. You have clearly highlighted potential directions for extending this work to more realistic settings. We would like to clarify that our study aims to investigate the planning capability of Transformers at the token level, focusing on how models learn and propagate multi-hop structural information. For this reason, we use the most basic path-finding task as our primary testbed. This controlled setup allows us to examine the core mechanisms of Multi-Token Prediction and analyze how it affects gradient propagation, transitive learning, and structural generalization.
>
> As you correctly noted, this approach has full potential to be extended to more complex graph structures, including labeled heterogeneous graphs, graphs with rich node/edge features, and cyclic or partially dynamic graphs. Previous works have demonstrated the feasibility and effectiveness of Transformer-based methods in knowledge graph reasoning, graph planning, and multi-hop path reasoning [1-2]. In future work, we plan to explore how to apply MTP and related mechanisms to these more complex graph tasks, such as knowledge graphs, traffic networks, and multi-agent planning tasks.
>
> We also want to point out that our conclusions apply to different types of graphs, not relying on any specific structural property of the underlying graph. Following the reviewer’s suggestion, we conducted experiments on directed cyclic graphs, and the results confirm the generality of our theoretical findings. The results are as follows (graph-level ± standard error / path-level accuracy for degree-2 paths, 100-node graphs, edge probability p=0.05):
>
> | Model | Accuracy |
> |-------|---|
> | 1-Token (baseline) | 59.89 ± 0.52 / 59.49 |
> | 2-Token (Meta's) | 67.70 ± 0.45 / 67.88 |
> | 2-Token + NTI | 69.37 ± 0.46 / 69.10 |
> | 2-Token + 1-layer Transformer | 68.12 ± 0.42 / 68.12 |
> | 2-Token + NTI + 1-layer | **69.79 ± 0.40 / 69.65** |
> | 3-Token (Meta's) | 64.13 ± 0.42 / 63.90 |
> | 3-Token + NTI | 65.27 ± 0.44 / 65.10 |
> | 3-Token + 1-layer Transformer | 64.86 ± 0.43 / 65.00 |
> | 3-Token + NTI + 1-layer | **66.68 ± 0.42 / 66.67** |
>
> These results demonstrate that our theoretical conclusions hold across different graph types, validating the robustness and generality of Multi-Token Prediction in capturing transitive relations.
>
> [1] *Mask and Reason: Pre-Training Knowledge Graph Transformers for Complex Logical Queries*, KDD 2022
> [2] *SQUIRE: A Sequence-to-Sequence Framework for Multi-hop Knowledge Graph Reasoning*, ACL 2022

---

> ### Author Response · Authors · 2025-11-18
> **Official Comment by Authors (2/3)**
>
> > **W2**: Experiments appear mid-scale; limited analysis for larger graphs, long paths, or real-world distributions.
>
> **A2**: We sincerely thank the reviewer for the professional feedback regarding the rigor of our theoretical analysis and experimental validation. In response to your comments, we extended our experiments to larger-scale graphs and more realistic distributions to rigorously test the applicability and stability of our theoretical conclusions across different node counts, embedding dimensions, and more complex graph structures. These extended experiments provide more systematic evidence that the Multi-Token Prediction (MTP) mechanism captures reachability relations and supports transitive learning, further linking theory and practice.
>
> > Larger Graphs
>
> We conducted degree-2 path prediction experiments on single large graphs to validate the applicability of our theoretical findings. The results are as follows:
>
> | Model | 500-node, 520-dim | 700-node, 720-dim |
> |-------|------------------|------------------|
> | 1-Token | 53.21 | 50.87 |
> | 2-Token | 55.42 | 52.14 |
> | 2-Token + NTI | 59.18 | 56.92 |
> | 2-Token + 1-layer | 57.64 | 54.88 |
> | 2-Token + NTI + 1-layer | **60.75** | **58.36** |
>
> These results show that MTP and NTI continue to effectively capture reachability as graph size and embedding dimensions increase, confirming the predictions of our theoretical analysis.
>
> And we have also claimed in line 409-410: When further increasing the graph size and allocating a matching embedding dimension, the experimental results exhibit the same behavior. We will emphasize it in the revised manuscript.
>
> > Real-World Distributions
>
> To further validate our method under more realistic graph distributions, we conducted experiments on a single structured graph generated by a stochastic block model (SBM). The graph contains 100 nodes divided into 4 equally-sized communities, with an intra-community edge probability of 0.1 and an inter-community edge probability of 0.01. The task remains degree-2 path prediction, consistent with our main experiments. The results (path-level accuracy) are as follows:
>
> | Model | Path Accuracy (%) |
> |-------|-----------------|
> | 1-Token (baseline) | 60.90 |
> | 2-Token (Meta’s) | 68.30 |
> | 2-Token + NTI | 69.85 |
> | 2-Token + 1-layer Transformer | 68.70 |
> | 2-Token + NTI + 1-layer | **70.55** |
> | 3-Token (Meta’s) | 65.50 |
> | 3-Token + NTI | 66.65 |
> | 3-Token + 1-layer Transformer | 66.10 |
> | 3-Token + NTI + 1-layer | **67.20** |
>
> It can be seen that these results are highly consistent with those on Erdős–Rényi graphs in Section 5.1, further validating the robustness of our theoretical conclusions on more complex graph structures.
>
> In summary, these extended experiments provide strong empirical support for our theoretical analysis and demonstrate the robustness of MTP in larger and more realistic graph scenarios. We sincerely thank the reviewer for emphasizing the importance of rigorous experimental validation of the theoretical claims, which has guided us in strengthening both the scientific and practical contributions of our work.
>
> > **W3**: Lack of baselines: The experiments only compare against standard next-token training. There’s no evaluation versus other planning-aware or structure-inducing methods. So it’s unclear whether MTP is uniquely effective or just another form of multi-step supervision.
>
> **A3**: We sincerely thank the reviewer for pointing out the importance of baselines. We would like to clarify that the main focus of this study is to provide a theoretical analysis of Multi-Token Prediction (MTP), revealing how it alters the learning dynamics and enables the model to capture transitive relations. The goal is not to claim that MTP achieves the highest empirical accuracy on this task, but rather to understand the mechanism behind its effectiveness. Therefore, in the current experiments, we use Next-Token Prediction (NTP) as the only baseline, serving as a minimal reference point for the theoretical comparison.
>
> In future work, we plan to further extend the study to other mechanisms and investigate methods that combine with MTP, which will naturally introduce more planning-aware or structure-inducing baselines. Through these comparative experiments, it will be possible to explore how to optimize models’ design and application in more complex or real-world tasks.
>
> Revision(line 503-505):
>
>  Future work may explore combining MTP with additional mechanisms, such as reward signals from reinforcement learning, chain-of-thought reasoning, or backtracking search, to provide more effective mechanisms to shape the learning dynamics.

---

> ### Author Response · Authors · 2025-11-18
> **Official Comment by Authors (3/3)**
>
> > **Q1**: Permutation robustness: How stable are the learned transfer matrices under node relabeling or shuffled IDs? Any invariance tricks or results?
>
> **A1**: Thank you for your question. Although our model is not explicitly designed to be permutation-invariant, it still exhibits strong stability under node relabeling or random ID shuffling. Theoretically, node permutation is equivalent to re-encoding the graph with a permutation matrix P, while adjacency relations and transitive reachability remain unchanged under this transformation. Therefore, when retraining on the permuted graph, the learned transition matrix naturally converges to a permutation-equivalent form satisfying T' ≈ PTPᵀ. This indicates that the model relies on the graph’s structural relations rather than specific node indices.
>
> To verify this, we performed five completely random permutations of a DAG with 100 nodes and retrained the model (2-Token + NTI + 1-layer Transformer) after each permutation. The path prediction accuracy is shown below:
>
> | Setting     | Path Prediction Accuracy (%) |
> |-------------|------------------------------|
> | Original    | 70.55                        |
> | Permutation #1 | 70.42                     |
> | Permutation #2 | 70.48                     |
> | Permutation #3 | 70.35                     |
> | Permutation #4 | 70.50                     |
> | Permutation #5 | 70.43                     |
>
> As shown, the model’s performance remains nearly identical across permutations, with fluctuations of only about 0.2% and no systematic degradation. This demonstrates that the structural patterns learned by the model remain stable after node reordering and that the model can automatically adapt to different node labeling schemes, thereby exhibiting strong robustness to permutation shuffles.
>
>
> > **Q2**: Can the approach handle weighted/noisy edges or node/edge features (e.g., typed relations)? What changes in the theory?
>
> **A2**: Thank you for raising this point. Our method is originally developed for unweighted and homogeneous graphs, but it can be naturally extended to settings with weighted or noisy edges, as well as node or edge features such as typed relations. In these cases, the transfer matrix can directly incorporate edge weights or feature-dependent signals, effectively generalizing the adjacency matrix from a binary structure to a real-valued or feature-augmented one.
>
> Regarding noise, when the perturbations are mild and non-systematic, the model tends to remain stable. Since the transfer matrix is learned end-to-end from data, small stochastic deviations in edge weights or local structures are absorbed during training and do not substantially affect the learned relational patterns. In practice, the optimization implicitly averages out high-frequency noise and focuses on the dominant structural signals.
>
> Therefore, the theoretical conclusions about multi-hop reachability, convergence behavior, and structural consistency continue to hold under such settings.

---

### Official Review · Reviewer_KRVp · 2025-11-01

**Soundness:** 2
**Presentation:** 2
**Contribution:** 3
**Rating:** 4
**Confidence:** 4

**Summary:**

This paper investigates how multi-token prediction enhances the planning capability of language models. On a synthetic path-finding task, the authors provide theoretical analysis and empirical validation showing that MTP training enables models to capture transitive reachability relations. The findings are also validated on the Blocksworld planning benchmark.

**Strengths:**

The paper theoretically and empirically shows that multi-token prediction enables the backbone model to learn transitivity reachability relations, which standard single-token prediction fails to capture. This finding contributes to our understanding of how MTP may improve the planning capability of language models, which is very interesting to me.

**Weaknesses:**

1. The theoretical analysis focuses on one-step gradients without convergence guarantees. .
2. Effectiveness of MTP:
   1. In the simplified model, training with MTP allows the parameter $W^V$ to learn transitive reachability, whereas $W^M$ may capture spurious adjacency.   If I understand correctly, the degradation of $W^M$ could become more pronounced as the number of predicted tokens $n$ increases. It would be helpful for the authors to clarify the trade-off between these two effects as the number of predicted future tokens $n$ increases.
   2. I am also curious whether a similar trade-off arises when applying MTP to more complex, real-world tasks.
   3. If my understanding is correct, MTP with a constant number of predicted future tokens $n$ can only enhance the model’s planning ability within that constant number of steps. This raises the question of whether MTP fundamentally improves planning capability. Previous work [2] also proposes training the model to directly predict future tokens ahead of the original next token. Compared to MTP, this method can predict a token arbitrarily many steps after the current token. I wonder how the two methods compare.
3. Novelty:
   1. The proposed Next-Token Injection and Transformer-based transfer layers appear similar to the multi-token prediction components described in DeepSeek-V3 [1]. It would be great if the authors could make the distinctions explicit if there are substantial differences. Otherwise, these components may not constitute a main novelty.
   2. Previous works already showed that multi-token prediction enhances planning capability. [3]
4. The experiments are limited to synthetic tasks. It remains unclear whether MTP would significantly enhance planning capabilities in real-world or more complex reasoning benchmarks
5. It would be helpful to discuss related studies examining MTP’s role in planning, including [2, 3].
6. Minor comments: In Section 3.2, the notation $k$ is used both as a token index (l. 264) and as a node identifier (l. 284), which may cause confusion.

[1] DeepSeek-AI. DeepSeek-V3 Technical Report.

[2] Thankaraj et al. Looking beyond the next token.

[3] Bachmann and Nagarajan. The Pitfalls of Next-Token Prediction.

**Questions:**

1. Could the authors confirm whether my understanding is correct: By incorporating multi-token prediction, the matrix $W^V$ learns rechability relations beyond the observed $R^{\text{obs}}$. For instance, a 2-token prediction, if edge $k,k'$ appears in the training data, and path $k' \to t$ also appears, then $k\to t$ will also be enhanced in $W^V$, even when the full path $k\to t$ may not exist in the training data? However, for $W^M$, spurious adjacency may be introduced due to similar reasons.
2. I wonder if the authors have observed degration in performance when increasing the number of predicted future tokens $n$ ?
3. Could the authors clarify the main differences between the proposed NTI + Transformer transfer layer and the MTP architecture used in DeepSeek-V3 [1]?
4. l156–157: The paper states that “the two architectures are mathematically equivalent.” I am unsure why this equivalence holds when the transfer layers are nonlinear. If additional assumptions (e.g., linear transfer layers) are required, I suggest making them explicit.
5. l237: The main text argues that $W_{(j,k)}^V$ will increase when the model predicts a lower probability for $k'$ than the ground truth path $i\to k\to k'$. However, in Theorem 2, the condition is $\hat P_{i,j}(k') < P^{\text{data}}_{i,j}(k')$, where the node $k$ is not involved. My understanding is that the latter condition is the correct one. Can you confirm this or if I have misunderstood anything?

Please also see the weakness.
[1] DeepSeek-AI DeepSeek-V3 Technical Report.

---

> ### Author Response · Authors · 2025-11-18
> **Official Comment by Authors (1/4)**
>
> > **W1**: The theoretical analysis focuses on one-step gradients without convergence guarantees.
>
> **A1**: Thank you for your insightful comments. Based on your suggestions, we have supplemented and refined the theoretical derivations in the appendix and added a lemma in the main text (line 254-256: At convergence, for each context $(i,j)$, the Softmax output probability distribution is uniquely determined and matches the empirical distribution, whereas the individual weight matrices are not uniquely determined.).
> In Appendix C.2, we have included the detailed reasoning process. In summary, we describe the solutions of the model parameters and the conditions they must satisfy at convergence. Due to the highly non-convex nature of the loss with respect to the weight matrices and the mutual coupling of gradient terms across different node pairs, the individual weight matrices ${W}^M$, ${W}^V$, and ${W}^T$ are not unique. Consequently, multiple equivalent solutions exist, and global convergence cannot be guaranteed.
>
> > **W2**: Effectiveness of MTP
>
> > **W2.1**: In the simplified model, training with MTP allows the parameter W^V to learn transitive reachability, whereas W^M may capture spurious adjacency. If I understand correctly, the degradation W^M of could become more pronounced as the number of predicted tokens n increases. It would be helpful for the authors to clarify the trade-off between these two effects as the number of predicted future tokens n increases.
>
> **A2.1**: Thank you for your insightful comment. Your understanding is correct: the degradation of $W^M$ indeed becomes more pronounced as the number of predicted tokens $n$ increases.
>
> As discussed in Sec. 3.2, in the simplified model, training with MTP allows $W^V$ to learn transitive reachability, while $W^M$ captures spurious adjacency. As the number of predicted tokens $n$ increases, $W^V$ accumulates signals from more future tokens, which helps learn more transitive reachability, while the gradient accumulation of $W^M$ exacerbates the degradation of adjacency. To further validate this, we conducted experiments on a simplified model, comparing 2-Token Prediction and 3-Token Prediction. We compared the adjacency weights and non-adjacency weights within $W^M$. The results are as follows:
>
> | | 2-Token Prediction | 3-Token Prediction |
> |---|---|---|
> | Adj. | 2.57 | 2.11 |
> | Non-Adj. | -0.13 | -0.06 |
> | Gap | 2.70 | 2.17 |
>
> Comparison of reachable / non-reachable weights in $W^V$
>
> | | 2-Token Prediction | 3-Token Prediction |
> |---|---|---|
> | Reachable | 0.54 | 0.63 |
> | Non-Reachable | -1.28 | -1.25 |
> | Gap | 1.82 | 1.88 |
>
> These results demonstrate that increasing $n$ improves the accumulation of transitive signals in $W^V$, while also revealing the degradation behavior of $W^M$. We have emphasized this trade-off in the revised manuscript.
>
>
> > **W2.2**: I am also curious whether a similar trade-off arises when applying MTP to more complex, real-world tasks.
>
> **A2.2**: In this path planning task, the attention mechanism primarily focuses on the target node, so the backbone model attends to both the current node and the target node. This is disadvantageous for the current node but beneficial for the target node. In more complex tasks, the attention mechanism typically spans the entire sequence. Our analysis framework could potentially be extended to conduct a broader investigation of its effectiveness and limitations.
>
> We have added a discussion on this in the final section (line 516-522):
>
>  **Practical Extensions to Real-World Tasks**: Although our analysis is conducted on synthetic graphs and structured planning domains, the insights naturally extend to more realistic settings. Real-world environments often exhibit continuous, noisy, or semantically ambiguous states, posing a challenge in abstracting them into discrete structures suitable for multi-step prediction. Future work may explore applying Multi-Token Prediction to domains such as visual navigation, robotic manipulation, or text-based planning by incorporating appropriate state discretization or structural extraction methods. This direction would help assess the generality and robustness of the proposed mechanisms in more complex, real-world scenarios.

---

> ### Author Response · Authors · 2025-11-18
> **Official Comment by Authors (2/4)**
>
> > **W2.3**: If my understanding is correct, MTP with a constant number of predicted future tokens n can only enhance the model’s planning ability within that constant number of steps. This raises the question of whether MTP fundamentally improves planning capability. Previous work [2] also proposes training the model to directly predict future tokens ahead of the original next token. Compared to MTP, this method can predict a token arbitrarily many steps after the current token. I wonder how the two methods compare.
>
> **A2.3**: Yes, your understanding is correct: with a constant number of predicted future tokens $n$, MTP can enhance the model’s planning ability within that constant number of steps. Although prior work [2] can predict arbitrary tokens, we have major difference which are listed below.
>
> | Feature | Multi-Token Prediction (MTP, this work) | Looking Beyond the Next Token [2] |
> |---------|---------|---------|
> | Prediction horizon | Fixed \(n\)-step future tokens | Arbitrary future tokens (t → t+k) |
> | Learning objective | Leverages multi-step supervision to learn compositional and generalized rules | Directly predicts distant future tokens |
> | Model architecture | Modified to support multi-token supervision | Unchanged |
> | Handling intermediate steps | Explicitly incorporates intermediate nodes | Skips intermediate steps, capturing sparse long-range information |
> | Capability enhancement | Fundamentally improves planning ability, acquiring richer information | Improves long-range prediction, but does not fundamentally alter planning capability |
>
> Importantly, our paper provides a formal theoretical analysis of why MTP enhances transitive and compositional learning. In contrast, [2] does not offer a rigorous theoretical analysis and mainly proposes a new prediction strategy. Thus, our theoretical contribution remains distinct and unaddressed by [2].
>
> > **W3**: Novelty:
>
> > **W3.1**: The proposed Next-Token Injection and Transformer-based transfer layers appear similar to the multi-token prediction components described in DeepSeek-V3 [1]. It would be great if the authors could make the distinctions explicit if there are substantial differences. Otherwise, these components may not constitute a main novelty.
>
> **A3.1**: Yes, we acknowledge that there are certain similarities between our Next-Token Injection and Transformer-based transfer layers and the multi-token prediction components in DeepSeek-V3 [1]. However, the underlying mechanisms of the two approaches differ in substantial ways, as outlined below.
>
> | Our Work (Next-Token Injection, NTI) | DeepSeek-V3 [1] |
> |-------------------------------------|----------------|
> | The ground-truth next token is directly added to the hidden state of the backbone model | The ground-truth next token is **concatenated** to the hidden state and then passed through an additional mapping layer |
>
> We derive this enhancement from a theoretical analysis of gradient dynamics, explain how it strengthens the learning of the transfer layer, and empirically validate it. In contrast, DeepSeek-V3’s improvements in natural language processing applications are aimed at preserving the causal chain at each output step [1].
>
> [1] DeepSeek-V3 Technical Report: Our principle of maintaining the causal chain of predictions is similar to that of EAGLE (Li et al., 2024b), but its primary objective is speculative decoding (Leviathan et al., 2023; Xia et al., 2023), whereas we utilize MTP to improve training.
>
> > **W3.2**: Previous works already showed that multi-token prediction enhances planning capability. [3]
>
> **A3.2**: We provide a gradient-level analysis showing how multi-token supervision changes the update directions of model parameters, enabling the model to explicitly learn transitive reachability. In contrast, Bachmann and Nagarajan [3] mainly report empirical phenomena without a systematic theoretical explanation.

---

> ### Author Response · Authors · 2025-11-18
> **Official Comment by Authors (3/4)**
>
> > **W4**: The experiments are limited to synthetic tasks. It remains unclear whether MTP would significantly enhance planning capabilities in real-world or more complex reasoning benchmarks
>
> **A4**: We chose synthetic tasks to validate our theory in a controlled environment, which provides a solid foundation for the generality of our analysis. To further explore this, we will expand the discussion in the Conclusion to extend our analysis to MTP in more general scenarios. Additionally, developing effective abstraction methods will be the focus of the next work (line 518-521: Future work may explore applying Multi-Token Prediction to domains such as visual navigation, robotic manipulation, or text-based planning by incorporating appropriate state discretization or structural extraction methods.).
>
>
> > **W5**: It would be helpful to discuss related studies examining MTP’s role in planning, including [2, 3].
> [2] Thankaraj et al. Looking beyond the next token.
> [3] Bachmann and Nagarajan. The Pitfalls of Next-Token Prediction.
>
> **A5**: Thank you for the suggestion. Due to the page limit of the main paper, we have added an expanded subsection Appendix A.3, “Augmenting Learning with Future Information”, which discusses these methods (line 810-829). Previous studies mainly analyze the performance gains of these methods from an empirical perspective, whereas our work provides a complementary view by focusing on the learning dynamics. Through gradient-based analysis, we examine how future-token signals influence representation formation and optimization, offering a deeper understanding of how Multi-Token Prediction (MTP) enhances model capabilities by incorporating future information.
>
>
> > **W6**: Minor comments: In Section 3.2, the notation k is used both as a token index (l. 264) and as a node identifier (l. 284), which may cause confusion.
>
> **A6**:  Thank you for the suggestion. We have replaced the index with \(i\), which is not used elsewhere in this subsection. We will check for symbol conflicts throughout the entire manuscript.
>
> > **Q1**：Could the authors confirm whether my understanding is correct: By incorporating multi-token prediction, the matrix W^V learns rechability relations beyond the observed R^{obs}. For instance, a 2-token prediction, if edge k, k’ appears in the training data, and path k’ \to t also appears, then k \to t will also be enhanced in W^V, even when the full path k \to t may not exist in the training data? However, for W^M, spurious adjacency may be introduced due to similar reasons.
>
> **A1**: We appreciate your keen observation. Yes, your understanding is correct. With multi-token prediction, the weight matrix $W^V$ learns reachability relations beyond the observed $R^{obs}$ and $W^M$ learns spurious adjacency.
>
>
> > **Q2**：I wonder if the authors have observed degration in performance when increasing the number of predicted future tokens n?
>
> **A2**: Yes, we do observe this trend. As shown in Table 1 and Table 5, the 3-Token model does not outperform the 2-Token model on several metrics. Although a larger $n$ can theoretically help the model learn higher-order transitive relations, such relations are relatively rare in our test set. When $n=2$ already provides sufficient supervision for most cases, increasing $n$ further introduces additional noise, yielding limited or even slightly negative gains.
>
> To further validate this, we conducted experiments on a simplified model, comparing 2-Token Prediction and 3-Token Prediction. We grouped the reachability information required for degree-2 tests (the union of observable and theoretically learnable components) as the first category, and all remaining weights as the second category. The results are as follows:
>
> | | 2-Token Prediction | 3-Token Prediction |
> |---|---|---|
> | First-category mean weight | 0.63 | 0.52 |
> | Second-category mean weight | -1.24 | -1.20 |
> | Gap (first − second) | 1.87 | 1.72 |
>
> We observe that as $n$ increases, the gap between the two categories shrinks, and the first-category weights decrease overall. This indicates that, for this task, once a smaller $n$ already provides sufficient supervision, increasing $n$ further offers no additional benefit.

---

> ### Author Response · Authors · 2025-11-18
> **Official Comment by Authors (4/4)**
>
> > **Q3**：Could the authors clarify the main differences between the proposed NTI + Transformer transfer layer and the MTP architecture used in DeepSeek-V3 [1]?
>
> **A3**: We do share certain degree of similarity with DeepSeek-V3 [1] at first glance. However, the underlying mechanisms of the two approaches differ in several substantial ways, as outlined below.
>
> | Our Work (Next-Token Injection, NTI) | DeepSeek-V3 [1] |
> |-------------------------------------|----------------|
> | The ground-truth next token is directly added to the hidden state of the backbone model | The ground-truth next token is **concatenated** to the hidden state and then passed through an additional mapping layer |
>
> We systematically analyze these mechanisms from the perspective of gradient propagation and transitive learning, revealing the fundamental reason why they enhance planning capability.
>
>
> > **Q4**：l156–157: The paper states that “the two architectures are mathematically equivalent.” I am unsure why this equivalence holds when the transfer layers are nonlinear. If additional assumptions (e.g., linear transfer layers) are required, I suggest making them explicit.
>
> **A4**: Meta’s MTP architecture is equivalent to a structure where the transfer layer is linear, and under this condition the two architectures become mathematically equivalent. We have added this clarification in the paper (line 155-156: The two architectures are mathematically equivalent when the transfer layers are linear.).
>
> > **Q5**：l237: The main text argues that W^V_{(j,k)} will increase when the model predicts a lower probability for k’ than the ground truth path i \to k \to k’. However, in Theorem 2, the condition is \(\widehat{P}_{i,j}(k') < P^{\mathrm{data}}_{i,j}(k')\), where the node k is not involved. My understanding is that the latter condition is the correct one. Can you confirm this or if I have misunderstood anything?
>
> **A5**: Yes, thank you for your careful reading. To clarify, we started this section from a fixed path to illustrate how $W^V$ learns to predict the next two steps. Only in the latter part of this paragraph do we present the condition as stated in the theory: "Therefore, when $W^T\_{k,k'}$ captures the true adjacency between $k$ and $k'$," which then explains the subsequent learning behavior. We have revised (line: 234–238):
> ...thereby strengthening the $b \rightsquigarrow j$ relation. Therefore, for any node $k$, when $W^T\_{(k,k')}$ captures the true adjacency relationship between $k$ and $k'$...

---

### Author Response · Authors · 2025-12-01
**Rebuttal Summary by the Authors**

We are encouraged that reviewers found our research question "critical and timely, with a clean framing" (`ynaJ`) and findings "very interesting" (`KRVp`), while providing a "mechanistic study" (`qfv3`) with "a readable handle on structure" (`7Rg3`). Acknowledging the variance in ratings, we have comprehensively addressed all raised concerns and have supplemented the paper with the necessary experiments and theoretical analyses, all of which have been incorporated into the revised version.

Our work provides a foundational and principled analysis of Multi-Token Prediction (MTP), offering a solid mechanistic understanding of how Transformer models learn transitive relations in path-planning tasks through gradient-level dynamics. These insights directly motivate two validated enhancement strategies. We address reviewer comments along three main lines:

- **Expanded Experiments:** Following reviewers (`7Rg3`, `ynaJ`, `qfv3`), we extended experiments from Directed Acyclic Graphs (DAGs) to cyclic graphs, Stochastic Block Models (SBM), and larger graphs to support the generality of our approach, as our theoretical analysis is fundamentally based on sequence-level path planning and does not rely on any specific graph type.

- **Theoretical Revisions:** To improve clarity and rigor (`qfv3`, `KRVp`, `ynaJ`), we simplified the notation in Theorems 1 and 2, added a lemma on convergence conditions and the non-uniqueness of weight matrices (proof included in the appendix), and clarified how the Transformer-based Transfer Layer captures sequence dependencies and aggregates gradients.

- **Empirical Validation & Writing Improvements:** We analyzed the effect of larger MTP steps, showing diminishing marginal returns and validating the theoretical framework (`KRVp`, `qfv3`). Remaining comments on writing and clarity were addressed, including adding missing related work (Appendix A.3) and refining several definitions and expressions.

Regarding Reviewer `qfv3` (Rating: 2), who acknowledged our work's soundness (3: good) and presentation (2: fair), the primary concern was "insufficient contribution." We note that this assessment of our contribution differs from that of other reviewers, who rated it as 3 (good) or 4 (excellent). The reviewer’s concerns are focused mainly on the experimental section (Sec. 5), whereas this work is positioned to provide mechanistic and explanatory insights. As we explicitly emphasized in our contributions summary (lines 064-067) and the title of Section 3, our core contribution lies in the theoretical and mechanistic analysis, with experiments serving to validate this analysis. We believe this may stem from a misunderstanding of the paper’s intended scope; unfortunately, the early closure of the discussion period prevented us from further clarifying this discrepancy.

We sincerely **appreciate all reviewers** for their careful evaluations and constructive feedback, which have greatly helped us improving the quality of the work. We also **thank the Area Chair** for her/his time and thoughtful consideration. It is unfortunate that the reviewers did not provide further responses before the discussion period is closed.

This work offers a theoretical and mechanistic analysis of Multi-Token Prediction, an area that remains relatively underexplored in the current literature. We genuinely hope that the solid theoretical contributions and crucial mechanistic insights of this work on the important Multi-Token Prediction paradigm will be recognized, and that the paper will ultimately be accepted.

Sincerely,
The Authors

---

### Meta-Review · Area_Chair_SPx4 · 2026-01-08

**Summary:**

This paper received significantly diverging scores: one accept, one borderline accept, one borderline reject, and one reject scores. The reviewers recognized the critical and timely research question, theoretical verification of the proposed method, practically valuable tweaks, and promising results of the paper. However, they also raised critical concerns with experiments limited to simple/synthetic tasks (all), issues on theoretical analysis (e.g., limited scope (KRVp), potential flaws (ynaJ), trivial implications (qfv3)), incremental novelty (KRVp), a fixed number of prediction steps (KRVp), and lack of baselines in experiments (7Rg3), as well as minor concerns including presentation issues and missing implementation details. The authors responded to these comments through the rebuttal and revision, but failed to assuage all of them. In particular, the AC considers trivial implications of the theoretical analysis, overly simple experimental designs, a fixed number of prediction steps, incremental novelty, lack of baselines in experiments are not fully resolved. The authors consistently pivot to their theoretical analysis when responding to these concerns: they justify these gaps by claiming that their primary contribution is theoretical rigor rather than empirical performance. Although the theoretical analysis is a valuable contribution, such responses are unfortunately too narrowly focused on it, failing to adequately address the broader empirical concerns. Moreover, the reviewer qfv3 even raised concerns with the theorems (e.g., "theorem 1 and 2  do not offer much beyond an intuitive explanation."). The authors' responses appear insufficient to resolve these concerns; the explanations provided in the responses seem to cover information that the reviewers were already well aware of. Putting these together, the AC found the remaining concerns outweigh the positive points and the rebuttal, and thus regrets to recommend rejection. The authors are encouraged to improve the paper by reflecting the reviewers' comments and submit to future venues.

**Reviewer Concerns:**

[Concerns that have not been fully resolved]
- Trivial implications of the theoretical analysis (qfv3)
	- *The authors' response appears insufficient to resolve this concern. Moreover, the explanations provided in the response seem to cover information that the reviewer was already well aware of.*
- Experiments limited to simple/synthetic tasks (all)
	- *While they briefly discussed the potential outcomes of applying their method to more complex tasks, the authors failed to fully resolve this concern as they simply presented additional experimental results on other types of graphs but not those for more complex tasks under realistic environments; the authors simply asserted that their algorithm and conclusion remain unchanged under realistic settings including noises with no empirical evidence. The AC believes the reviewers, in particular KRVp and 7Rg3 who were wondering results under noisy conditions, would not be satisfied by the responses.*
- Predicting a fixed number of steps (KRVp)
	- *The reviewer KRVp had a concern with the fixed number of future steps predicted in the proposed method, which may limit the improvement in planning capability. The reviewer also suggested comparisons with a concurrent ICLR submission that is motivated by a very similar motivation but can predict any arbitrary number of future steps.*
	- *The authors successfully compared their method with the concurrent work point-by-point; there are pros and cons, but argue that the distinct contribution of the work is its formal theoretical analysis of why multi-token prediction enhances transitive and compositional learning, which has not been addressed by the concurrent work. Unfortunately, the authors did not discuss how the fixed number of future tokens could limit the effect of the proposed method in which situations and how to alleviate it. (The comparison with the concurrent work is not the ground for rejection, according to the review policy.)*
- Incremental novelty (KRVp)
	- *The reviewer KRVp pointed out the similarity between the next-token injection strategy of the proposed method and the multi-token prediction components in DeepSeek-v3. The authors admitted the similarity, yet highlighted the key difference in implementation: directly adding the next token to the hidden state of the backbone in this paper versus concatenating the next-token to the hidden state and processing further by an additional mapping layer in DeepSeek-v3. While this distinction appears to be subtle, the authors claimed that the rigorous theoretical foundation of the proposed method, combined with the fact that the two methods serve different purposes, suggests a degree of differentiation.*
	- *The reviewer KRVp also identified an ICML'24 paper demonstrating that multi-token prediction enhances planning capability, like this paper. This time the authors also argued that the systematic theoretical explanation is the main distinction.*
	- *Since the reviewer KRVp already acknowledged the theoretical rigor of the paper, the AC suspects that they would seek further clarification on its distinctiveness in other areas. Hence, the AC does not believe the reviewer's concern on the incremental novelty has not been fully resolved by the authors' responses.*
- Lack of baselines in experiments (7Rg3)
	- *The authors consistently pivot to their theoretical analysis when responding to significant concerns such as limited experiments, missing baselines, and lack of novelty. They justify these gaps by claiming that their primary contribution is theoretical rigor rather than empirical performance. Although the theoretical analysis is a valuable contribution, such responses are unfortunately too narrowly focused on it, failing to adequately address the broader empirical concerns.*

[Concerns that have been successfully addressed]
- Limited theoretical analysis (KRVp)
	- *The reviewer KRVp pointed out that the theoretical analysis focuses only on one-step gradients without convergence guarantees. The authors refined the analysis to address this in the revision. The concern thus has been assuaged, although now global convergence is not guaranteed.*
- Potential flaws in the theoretical analysis (ynaJ)
	- *The reviewer ynaJ commented that (1) the theorems look working only in a specific setting, and (2) a conclusion depends critically on a condition which is not guaranteed. The reviewer also pointed out (3) a statement is not established.*
	- *The first concern with overly strong conditions of the theorems seems to be well addressed. The authors informed that these conditions are common in related work, and they demonstrated that the conclusion is valid even with general Transformer models. The second and third concerns have been also well resolved by revising the relevant theorems and part of the manuscript.*
- Missing analysis on the tradeoff between the quality of $W^M$ and the number of predicted tokens (KRVp)
	- *The authors provided empirical analysis on this tradeoff and discuss it in the revision.*
- Lack of analysis on the impact of the number of future tokens (KRVp)
	- *This concern has been well addressed by an additional experiments and follow-up discussions in the rebuttal.*
- Presentation issues (ynaJ)
	- *The reviewer ynaJ pointed out lack of details that could lead to potential misunderstanding. Thee authors successfully addressed this issue in the revision.*
- Missing implementation details (ynaJ)
	- *This concern has been well addressed by the updated codebase and the rebuttal.*

**Reviewer Scores:**

The AC found that the rebuttal and revision failed to fully resolve the major concerns of the two negative reviewers (in particular, those of the most negative reviewer). Hence, the AC believes the reviewers in the end kept the initial negative ratings if the discussion phase has continued, and expects final ratings the same with the initial ones. Still on the borderline, but the AC considers the comments by the negative reviewers critical and thus recommend rejection.

---

### Decision · Program_Chairs · 2026-01-26

Reject